# Geometry of the Loss Landscape in Invariant Deep Linear Neural Networks

## Abstract

Equivariant and invariant machine learning models seek to take advantage of symmetries and other structures present in the data to reduce the sample complexity of learning. Empirical work has suggested that data-driven methods, such as regularization and data augmentation, may achieve a comparable performance as genuinely invariant models, but theoretical results are still limited. In this work, we conduct a theoretical comparison of three different approaches to achieve invariance: data augmentation, regularization, and hard-wiring. We focus on mean squared error regression with deep linear networks, where we specifically consider rank-bounded linear maps which do not have a linear parametrization and which can be hard-wired to be invariant to specific group actions. We show that the optimization problems resulting from hard-wiring and data augmentation have the same critical points, all of which are saddles except for the global optimum. In contrast, regularization leads to a larger number of critical points, again all of which are saddles except for the global optimum. The regularization path is continuous and converges to the optimum of the hard-wired problem.

## 1 Introduction

Equivariant and invariant models are a class of machine learning models designed to incorporate specific symmetries or invariances that are known to exist in the data. An equivariant model ensures that when the input undergoes a certain transformation, the model's output transforms in a predictable way. Many powerful hard-wired equivariant and invariant structures have been proposed over the recent years (see, e.g., Cohen & Welling, 2016; Zaheer et al., 2017; Geiger & Smidt, 2022; Liao et al., 2024). Such models are widely employed and have achieved state-of-the-art level performance across various scientific fields, including condensed-matter physics (Fang et al., 2023), catalyst design (Zitnick et al., 2020), drug discovery (Igashov et al., 2024), as well as several others.

Given an explicit description of the desired invariance and equivariance structures, a direct way to implement them is by hard-wiring a neural network in a way that constraints the types of functions that it can represent so that they are contained within the desired class. Another intuitive method to approximately enforce invariance and equivariance is data augmentation, where one instead supplies additional data in order to guide the network towards selecting functions from the desired class. Both approaches have shown to be viable for obtaining invariant or equivariant solutions (see, e.g., Gerken & Kessel, 2024; Moskalev et al., 2023). However, it is not entirely clear how the learning processes and in particular the optimization problems compare. An obvious drawback of data augmentation is that the number of model parameters as well as the number of training data points may be large. On the other hand, it is known that constrained models (Finzi et al., 2021), or underparameterized models, can have a more complex optimization landscape, but the specific interplay between the amount of data and the structure of the data is not well understood. We are interested in the following question: how do invariance, regularization, and data augmentation influence the optimization process and the resulting solutions of learning? To start developing an understanding, we investigate the simplified setting of invariant linear networks, for which we investigate the static loss landscape of the three respective optimization problems.

The loss landscapes of neural networks are among the most intriguing and actively studied topics in theoretical deep learning. In particular, a series of works has documented the benefits of overparameterization in making the optimization landscape more benevolent (see, e.g., Poston et al., 1991;

Gori & Tesi, 1992; Soltanolkotabi et al., 2019; Simsek et al., 2021; 2023; Karhadkar et al., 2024). This stands at odds with the success of data augmentation, since when using data augmentation as done in practice, even enormous models may no longer be overparameterized and may have fewer parameters than the number of training data points (see, e.g., Garg et al., 2022; Belkin et al., 2019). Beyond overparameterization, the effects of different architecture choices on the loss landscape are of interest (see, e.g., Li et al., 2018). As mentioned above, equivariant and invariant architectures are of particular interest, as they could potentially help dramatically reduce the sample complexity of learning within a clearly defined framework. This has been documented theoretically in a recent stream of works (see, e.g., Mei et al., 2021; Tahmasebi & Jegelka, 2023). However, the impact of these architecture choices on the optimization landscape is still underexplored. Equivariant linear networks have received interest as simplified models to obtain concrete and actionable insights for more complex neural networks (see, e.g., Chen & Zhu, 2023; Kohn et al., 2022; Zhao et al., 2023; Nordenfors et al., 2024).

Our work advances this line of investigation by considering the optimization problems arising from data augmentation, regularization and hard-wiring. We consider linear networks whose end-to-end functions are rank-constrained and thus cannot be simply re-parameterized as linear models. The non-convexity of the function space makes it nontrivial to draw conclusions about the impact of the constraints imposed by invariances. These models are a natural point of departure to study other networks with nonlinear function space, such as networks with nonlinear activation functions. We observe in particular that rank constraints are common in practice. For example, in generative models such as variational autoencoders (VAEs) (Kingma & Welling, 2022), the hidden layer is usually narrower than the input and output layers with the purpose of capturing a low-dimensional latent representation of the data. In large language models, low-rank adaptation (LoRA) (Hu et al., 2021) is also used to reduce the number of trainable parameters for downstream tasks. In these and other cases where the architecture has a narrow intermediate linear layer, rank constraints arise.

## 1.1 CONTRIBUTIONS

In this work, we study the impact of invariance in learning by considering and comparing the optimization problems that arise in linear invariant neural networks with a non-linear function space.

- We consider three optimization problems: data augmentation, constrained model, and regularization. We show that these problems are equivalent in terms of their global optima, in the limit of strong regularization and full data augmentation.

- We study the regularization path and show that it continuously connects the global optima of the regularized problem and the global optima of the constrained invariant model.

- We are able to characterize all the critical points in function space for all three problems. In fact, the critical points for data augmentation and the constrained model are the same. There are more critical points for the unconstrained model with regularization.

## 1.2 RELATED WORK

**Loss Landscapes** The static optimization landscape of linear networks has been studied in numerous works, whereby most works consider fully-connected networks. In particular, the seminal work of Baldi & Hornik (1989) showed for a two-layer linear network that the square loss has a single minimum up to trivial symmetry and all other critical points are saddles. Kawaguchi (2016) considered the deep case and showed the existence of bad saddles in parameter space for networks with three or more layers. Laurent & Brecht (2018) showed that for deep linear networks with no bottlenecks, all local minima are global for arbitrary convex differentiable losses, and Zhou & Liang (2018) offered a full characterization of the critical points for the square loss. The more recent work of Trager et al. (2020) found that for deep linear networks with bottlenecks, the non-existence of non-global local minima is very particular to the square loss. Several works have also considered more specialized linear network architectures, such as symmetric parametrization (Tarmoun et al., 2021) and deep linear convolutional networks (Kohn et al., 2022; 2024a). These and the recent work of Shahverdi (2024) also discuss the critical points in parameter and in function space. In this context we may also highlight the work of Levin et al. (2024), which studies the effect of parametrization on an optimization landscape. In contrast to these works, we focus on deep linear networks with bottle-

necks that are invariant to a given group action. The corresponding functions are rank-constrained and thus cannot be simply re-parameterized as linear models.

**Training Dynamics** Although analyzing the training dynamics is not the main focus of our work, we would like to briefly highlight several related works in this direction. Many works have studied the convergence of parameter optimization in deep linear networks, which to this date remains an interesting topic even in the case of fully-connected layers (Arora et al., 2018; 2019a;b; Xu et al., 2023; Bah et al., 2021; Bréchet et al., 2023; Saxe et al., 2013). For certain types of linear convolutional networks, Gunasekar et al. (2018) studied the implicit bias of parameter optimization. In the context of equivariant models, Chen & Zhu (2023) discuss the implicit bias of gradient flow on linear equivariant steerable networks in group-invariant binary classification.

**Invariance, regularization, and data augmentation** A few works try to understand the difference between the various aforementioned methods to achieve invariance. Geiping et al. (2023) seek to disentangle the mechanisms through which data augmentation operates and suggest that data augmentation that promotes invariances may provide greater value than enforcing invariance alone, particularly when working with small to medium-sized datasets. Beside data augmentation, Botev et al. (2022) claims that explicit regularization can improve generalization and outperform models that achieve invariance by averaging predictions of non-invariant models. Moskalev et al. (2023) empirically show that the invariance learned by data augmentation deteriorates rapidly, while models with regularization maintain low invariance error even under substantial distribution drift. Our work is inspired by their experiments, and we seek to theoretically study whether data augmentation can learn genuine invariance. A recent work by Kohn et al. (2024b) investigates linear neural networks through the lens of algebraic geometry and computes the dimension, singular points, and the Euclidean distance degree, which serves as an upper bound on the complexity of the optimization problem. We are also consider the number of critical points but are primarily interested in the comparison of the loss landscapes arising from different methods. The work of Gideoni (2023) investigates the training dynamics of linear regression with data augmentation. In contrast, we consider regression with rank-bounded linear maps and also discuss the effect of regularization. Nordenfors et al. (2024) investigate the optimization dynamics of a neural network with data augmentation and compare it to an invariance hard-wired model. The authors show that the data augmented model and the hard-wired model have the same stationary points within the set of representable equivariant maps $\mathcal{E}$, but does not offer conclusions about stationary points that are not in $\mathcal{E}$. In contrast, we obtain a result that describes all critical points in a non-linear function space of rank-constrained linear maps and show that all of them are indeed invariant.

## 2 PRELIMINARIES

We use $[n]$ to denote the set $\{1, 2, \ldots, n\}$. $\mathbf{I}_d$ represents a $d$ by $d$ identity matrix. For any square matrix $U \in \mathbb{C}^{n \times n}$, we use $U_r \in \mathbb{C}^{n \times r}$ to denote the truncation of $U$ to its first $r$ columns. In a slight abuse of notation, for any non-square matrix $\Sigma \in \mathbb{C}^{n \times m}$, we use $\Sigma_r \in \mathbb{C}^{r \times r}$ to denote the truncation of $\Sigma$ to its first $r$ columns and $r$ rows. For any matrix $M \in \mathbb{C}^{n \times m}$, we denote the Hermitian as $M^\dagger$, the Moore-Penrose pseudoinverse as $M^+$, and the transpose as $M^\mathrm{T}$. We use $\|M\|_2$ and $\|M\|_F$ to denote the operator norm and the Frobenius norm of $M$, respectively. For a matrix $M \in \mathbb{R}^{n \times m}$, we use $\mathrm{vec}(M)$ to denote the column by column vectorization of $M$ in $\mathbb{R}^{nm}$. Given any two vector spaces $V$ and $W$, we use $V \otimes W$ to denote the tensor (Kronecker) product of $V$ and $W$.

### 2.1 EQUIVARIANCE AND INVARIANCE

To set up our problem, we need to borrow some concepts from representation theory.

**Definition 1.** A *representation* of a group $\mathcal{G}$ on a vector space $\mathcal{X}$ is a group homomorphism $\rho \colon \mathcal{G} \to GL(\mathcal{X})$, where $GL(\mathcal{X})$ is the group of invertible linear transformations on $\mathcal{X}$.

**Definition 2.** Let $\mathcal{X}$ and $\mathcal{Y}$ be two vector spaces with representations $\rho_\mathcal{X}$ and $\rho_\mathcal{Y}$ of the same group $\mathcal{G}$, respectively. A function $f \colon \mathcal{X} \to \mathcal{Y}$ is said to be *equivariant* with respect to $\rho_\mathcal{X}$ and $\rho_\mathcal{Y}$ if

$$f \circ \rho_\mathcal{X}(g) = \rho_\mathcal{Y}(g) \circ f, \quad \forall g \in \mathcal{G}. \tag{1}$$

If $f$ is a linear function, we say $f$ is a $\mathcal{G}$-*linear map* or a $\mathcal{G}$-*intertwiner*. For simplicity of notation, we write $f(gx) = gf(x)$ when $\rho_\mathcal{X}$ and $\rho_\mathcal{Y}$ are clear. If $\rho_\mathcal{Y}$ is the trivial representation, i.e., $\rho_\mathcal{Y}(g)$

is the identity map for all $g \in \mathcal{G}$, then $f$ is said to be *invariant* with respect to $\rho_{\mathcal{X}}$. We then write $f(gx) = f(x)$ when $\rho_{\mathcal{X}}$ is clear.

For a finite cyclic group $\mathcal{G}$ there is a generator $g \in \mathcal{G}$ such that $\mathcal{G} = \{e, g, g^2, \dots, g^{n-1}\}$, where $e$ is the identity element, $n$ is the order of the group, and $g^i = g^j$ whenever $i \equiv j \mod n$.

**Example 1.** *For example, the rotational symmetries of a polygon with $n$ sides in $\mathbb{R}^2$ form a group. The group is a cyclic group of order $n$, i.e., $\mathcal{G} = C_n$ with generator $g$, and the representation is generated by $\rho(g) = \begin{bmatrix} \cos\frac{\pi}{n} & -\sin\frac{\pi}{n} \\ \sin\frac{\pi}{n} & \cos\frac{\pi}{n} \end{bmatrix}$.*

### 2.2 DEEP LINEAR NEURAL NETWORKS

A *linear neural network* $\Phi(\boldsymbol{\theta}, \mathbf{x})$ with $L$ layers of widths $d_1, \dots, d_L$ is a model of linear functions

$$\Phi(\boldsymbol{\theta}, \mathbf{x}) : \mathbb{R}^{d_{\boldsymbol{\theta}}} \times \mathbb{R}^{d_{\mathcal{X}}} \to \mathbb{R}^{d_{\mathcal{Y}}}; \quad \mathbf{x} \mapsto W_L \cdots W_1 \mathbf{x}, \tag{2}$$

parameterized by weight matrices $W_j \in \mathbb{R}^{d_j \times d_{j-1}}, \forall j \in [L]$. We write $\boldsymbol{\theta} = (W_L, \dots, W_1) \in \Theta \subseteq \mathbb{R}^{d_{\boldsymbol{\theta}}}$ for the tuple of weight matrices. The dimension of the *parameter space* $\Theta$ is $d_{\boldsymbol{\theta}} = \sum_{j \in [L]} d_j d_{j-1}$, where $d_0 := d_{\mathcal{X}}$, $d_L := d_{\mathcal{Y}}$ are the input and output dimensions, respectively.

For simplicity of the notation, we will write $W := W_L \cdots W_1$ for the end-to-end matrix, and write $W_{j:i} := W_j \cdots W_i$ for the matrix product of layer $i$ up to $j$ for $1 \leqslant i \leqslant j \leqslant L$. We denote the network's parameterization map by

$$\mu : \Theta \to \mathbb{R}^{d_L \times d_0}; \quad \boldsymbol{\theta} = (W_1, \dots, W_N) \mapsto W = W_N \cdots W_1. \tag{3}$$

The network's *function space* is the image of the parametrization map $\mu$, which is the set of linear functions it can represent, i.e., the set of $d_L \times d_0$ matrices of rank at most $r := \min\{d_0, \dots, d_L\}$. We denote the function space by $\mathcal{M}_r \subseteq R^{d_L \times d_0}$. When $r = \min\{d_0, d_L\}$, the function space is a vector space which can represent any linear function mapping from $\mathbb{R}^{d_0}$ to $\mathbb{R}^{d_L}$. On the other hand, when $r < \min\{d_0, d_L\}$, it is a non-convex subset of $\mathbb{R}^{d_L \times d_0}$, known as a *determinantal variety* (see Harris, 1992, Chapter 9), which is determined by polynomial constraints, namely the vanishing of the $(r+1) \times (r+1)$ minors. We adopt the following terminology from Trager et al. (2020).

**Definition 3.** The parametrization map $\mu$ is *filling* if $r = \min\{d_0, d_L\}$. If $r < \min\{d_0, d_L\}$, then $\mu$ is *non-filling*. In the filling case, $\mathcal{M}_r = \mathbb{R}^{d_L \times d_0}$, which is convex. In the non-filling case, $\mathcal{M}_r \subsetneq R^{d_L \times d_0}$ is a determinantal variety, which is non-convex.

Given a group $\mathcal{G}$, a representation $\rho_{\mathcal{X}}$ on the input space $\mathcal{X}$ and a representation $\rho_{\mathcal{Y}}$ on the output space, an *equivariant linear network* is a linear neural network $\Phi(\boldsymbol{\theta}, \mathbf{x})$ that is equivariant with respect to $\rho$, i.e., $W_L \cdots W_1 \rho_{\mathcal{X}}(g)x = \rho_{\mathcal{Y}}(g) W_L \cdots W_1 x$ for all $g \in \mathcal{G}$ and $x \in \mathcal{X}$. When $\rho_{\mathcal{Y}}$ is trivial, the network is called an *invariant linear network*. Though we focus on invariant linear networks, it is easy to extend all the results to equivariant linear networks by constructing a new representation taking the tensor product of $\rho_{\mathcal{X}}$ and $\rho_{\mathcal{Y}}$ (see Appendix A.2). In section 4 we will discuss how to define a deep linear network that is hard-wired to be invariant to a given group.

### 2.3 LOW RANK APPROXIMATION

For a linear network with $r = \min\{d_0, \dots, d_L\}$, the function space consists of $d_L \times d_0$ matrices of rank at most $r$. Optimization in such a model is closely related to the problem of approximating a given matrix by a rank bounded matrix. When the approximation error is measured in Frobenius norm, Eckart & Young (1936a) show that the optimal bounded-rank approximation of a matrix is given in terms of the top components in its singular value decomposition (see, e.g., Strang, 2019, I.9): If $A = U\Sigma V^{\mathrm{T}} = \sigma_1 u_1 v_1^{\mathrm{T}} + \cdots + \sigma_n u_n v_n^{\mathrm{T}}$ and $B$ is any matrix of rank $r$, then $\|A - B\|_F \geq \|A - A_r\|_F$, where $A_r = \sigma_1 u_1 v_1^{\mathrm{T}} + \cdots + \sigma_r u_r v_r^{\mathrm{T}}$. Mirsky (1960) showed that this result in fact holds for any matrix norm that depends only on the singular values.

There are several generalizations of this result, for instance to bounded-rank approximation with some fixed entries (Golub et al., 1987), weighted least squares (Ruben & Zamir, 1979; Dutta & Li, 2017), and approximation of symmetric matrices by rank-bounded symmetric positive semidefinite matrices (Dax, 2014). However, for general norms or general matrix constraints, the problem is

known to be hard (Song et al., 2017; Gillis & Shitov, 2019). We will be interested in the problem of approximating a given matrix with a rank-bounded matrix that is constrained to within the set of matrices that represent linear maps that are invariant to a given group.

## 3 MAIN RESULTS

### 3.1 GLOBAL OPTIMUM IN CONSTRAINED FUNCTION SPACE

As we want our function space to contain only the $\mathcal{G}$-interwiners, we need to constrain it accordingly. Due to the linearity of the representation $\rho_{\mathcal{X}}$, the constraints are also linear in $\mathbb{R}^{d_L \times d_0}$. Prior research has investigated the constraints for different groups (see, e.g., Maron et al., 2019; Puny et al., 2023; Finzi et al., 2021). We have the following proposition to explicitly characterize the constraints, proved in Appendix A.1. We will focus on the case where the group $\mathcal{G}$ is finite and cyclic, the representation $\rho_{\mathcal{X}}$ is given and nontrivial, and the representation $\rho_{\mathcal{Y}}$ is trivial.

**Proposition 1.** *Given a cyclic group $\mathcal{G}$ and a representation $\rho_{\mathcal{X}}$ of $\mathcal{G}$ on vector space $\mathcal{X} = \mathbb{R}^{d_0}$, a linear function $W$ mapping from $\mathcal{X}$ to $\mathcal{Y} = \mathbb{R}^{d_L}$ is invariant with respect to $\rho_{\mathcal{X}}$ if and only if $WG = 0$, where $G = \mathbf{I}_{d_0} - \rho_{\mathcal{X}}(g)$, and $g$ is the generator of $\mathcal{G}$.*

**Remark 1.** *Though we assume that $\mathcal{G}$ is cyclic, the above proposition can be generalized to any finitely generated group $\mathcal{G}$ by replacing the single generator $g$ with a set of generators $\{g_1, \ldots, g_M\}$. For that, define $G_m = \mathbf{I}_{d_0} - \rho_{\mathcal{X}}(g_m)$ for all $m \in [M]$, and set $G = [G_1, \ldots, G_M]$ a $d_0 \times (Md_0)$ matrix. In fact, we can even extend this proposition to continuous groups such as Lie groups. As discussed by Finzi et al. (2021), for any Lie group $\mathcal{G}$ of dimension $M$ with its corresponding Lie algebra $\mathfrak{g}$, we are able to find a basis $\{A_1, \ldots, A_M\}$ for $\mathfrak{g}$. If the exponential map is surjective in $\mathcal{G}$, we can then use it to parameterize all elements in $\mathcal{G}$, i.e., for any $g \in \mathcal{G}$, we can find weights $\{\alpha_m \in \mathbb{R}\}_{m \in [M]}$ such that $g = \exp\left(\sum_{m=1}^{M} \alpha_m A_m\right)$. Therefore, $G_m = d\rho_{\mathcal{X}}(A_m)$ and $G = [G_1, \ldots, G_M]$, where $d\rho$ is the Lie algebra representation. See Appendix A.1 for more details.*

Consider a data set $\mathcal{D} = \{(\mathbf{x}_i, \mathbf{y}_i)\}_{i=1}^{n}$, a cyclic group $\mathcal{G}$, and a representation $\rho_{\mathcal{X}}$ of $\mathcal{G}$ on vector space $\mathcal{X} = \mathbb{R}^{d_0}$. Let $X = \{\mathbf{x}_1, \ldots, \mathbf{x}_n\} \in \mathbb{R}^{d_0 \times n}$, $Y = \{\mathbf{y}_1, \ldots, \mathbf{y}_n\} \in \mathbb{R}^{d_L \times n}$. Given a positive integer $r < \min\{d_0, d_L\}$, we want to find an invariant linear and rank-bounded function that minimizes the empirical risk, i.e., we want to solve the following optimization problem:

$$\widehat{W} = \underset{W \in \mathbb{R}^{d_L \times d_0}}{\arg\min} \frac{1}{n} \|WX - Y\|_F^2, \quad \text{s.t.} \quad WG = 0, \ \text{rank}(W) \leq r. \tag{4}$$

We assume $XX^{\mathrm{T}}$ has full rank $d_0$ such that we can use its positive definite square root $P = (XX^{\mathrm{T}})^{1/2} \in \mathbb{R}^{d_0 \times d_0}$ to derive:

$$\begin{aligned}
\|WX - Y\|_F^2 &= \langle WX, WX \rangle_F - 2\langle WX, Y \rangle_F + \langle Y, Y \rangle_F \\
&= \langle WP, WP \rangle_F - 2\langle WP, YX^{\mathrm{T}}P^{-1} \rangle_F + \langle Y, Y \rangle_F \\
&= \|WP - YX^{\mathrm{T}}P^{-1}\|_F^2 + \text{const} \\
&= \|\widetilde{W} - YX^{\mathrm{T}}P^{-1}\|_F^2 + \text{const}, \quad \text{where } \widetilde{W} = WP. \tag{5}
\end{aligned}$$

We can see that the above optimization problem (4) is equivalent to the following low-rank approximation problem:

$$\widehat{\widetilde{W}} = \underset{\widetilde{W} \in \mathbb{R}^{d_L \times d_0}}{\arg\min} \frac{1}{n} \|\widetilde{W} - Z\|_F^2, \quad \text{s.t.} \quad \widetilde{W}\widetilde{G} = 0, \ \text{rank}(\widetilde{W}) \leq r, \tag{6}$$

where $Z = YX^{\mathrm{T}}P^{-1}$ and $\widetilde{G} = P^{-1}G$. If we get the solution $\widehat{\widetilde{W}}$, then we can recover the solution to (4) by $\widehat{W} = \widehat{\widetilde{W}}P^{-1}$. Since $\widetilde{W}\widetilde{G} = 0$, we know that the rows of $\widetilde{W}$ are in the left null space of $\widetilde{G}$. Then $\text{rank}(\widetilde{W}) \leq \text{nullity}(\widetilde{G}) = d_0 - \text{rank}(\widetilde{G})$. In order to make this low rank constraints nontrivial, we suppose $r < d := \text{nullity}(\widetilde{G})$. In the case where $r \geq d$, the projection of the unique least square estimator onto the left null space already satisfies the rank constraint, making the rank constraint meaningless. The following theorem characterizes the solution to the above optimization problem, proved in Appendix A.3.

**Theorem 1.** *Denote* $\overline{Z}^{inv} := Z(\mathbf{I}_{d_0} - \widetilde{G}\widetilde{G}^{+})$. *We assume* $\mathrm{rank}(\overline{Z}^{inv}) > r$. *Let* $\overline{Z}^{inv} = \overline{U}^{inv}\overline{\Sigma}^{inv}\overline{V}^{inv\mathrm{T}}$ *be the SVD of* $\overline{Z}^{inv}$. *Then the solution to (4) is* $\widehat{W}^{inv} = \overline{U}_r^{inv}\overline{\Sigma}_r^{inv}\overline{V}_r^{inv\mathrm{T}}P^{-1}$.

**Remark 2.** *The assumption that* $\mathrm{rank}(\overline{Z}^{inv}) > r$ *is mild. Fix any full row rank data matrix* $X$ *and suppose* $Y = WX + E$, *where* $E \in \mathbb{R}^{d_L \times n}$ *is a random noise matrix. If each column of* $E$ *is drawn independently from any continuous distribution with full support on* $\mathbb{R}^{d_L}$, *then with probability* 1, $\mathrm{rank}(\overline{Z}^{inv}) = \min\{d, d_L, d_0\} > r$. *In Appendix A.9 we verified this on the MNIST data set.*

The key observation is that if the target matrix lives in the invariant linear subspace, then the low-rank approximator of that matrix also lives in the invariant linear subspace. Theorem 1 shows how to find the global optima in the optimization problem of constrained space. Indeed, we can project the target matrix to the left null space of $\widetilde{G}$ and find its low-rank approximator.

## 3.2 GLOBAL OPTIMUM IN FUNCTION SPACE WITH REGULARIZATION

Instead of imposing constraints on the function space, we can also regularize the optimization problem. We consider the following optimization problem:

$$\widehat{W} = \underset{W \in \mathbb{R}^{d_L \times d_0}}{\arg\min} \frac{1}{n}\|WX - Y\|_F^2 + \lambda\|WG\|_F^2, \quad \text{s.t.} \quad \mathrm{rank}(W) \leq r. \tag{7}$$

Similarly to optimization problem (4), we can rewrite problem (7) in the following form:

$$\widehat{\widetilde{W}} = \underset{\widetilde{W} \in \mathbb{R}^{d_L \times d_0}}{\arg\min} \frac{1}{n}\|\widetilde{W} - Z\|_F^2 + \lambda\|\widetilde{W}\widetilde{G}\|_F^2, \quad \text{s.t.} \quad \mathrm{rank}(\widetilde{W}) \leq r. \tag{8}$$

The optimization problem (8) is referred to as *manifold regularization* (Zhang & Zhao, 2013). In the context of manifold regularization, the input data points are assumed to lie on a low-dimensional manifold embedded in a high-dimensional space. The following proposition, characterizing the solution to the above optimization problem, can be established directly by following the manifold regularization result of Zhang & Zhao (2013, Theorem 1).

**Proposition 2.** *Denote* $B(\lambda)$ *the square root of the symmetric positive definite matrix* $\mathbf{I}_{d_0} + n\lambda\widetilde{G}\widetilde{G}^{\mathrm{T}}$, *i.e.*, $B(\lambda)^2 = \mathbf{I}_{d_0} + n\lambda\widetilde{G}\widetilde{G}^{\mathrm{T}}$. *Denote* $\overline{Z(\lambda)}^{reg} = ZB(\lambda)^{-1}$, *and* $\overline{Z(\lambda)}^{reg} = \overline{U(\lambda)}^{reg}\overline{\Sigma(\lambda)}^{reg}\overline{V(\lambda)}^{reg\mathrm{T}}$ *as the SVD of* $\overline{Z(\lambda)}^{reg}$. *Then the solution to problem 7 is* $\widehat{W(\lambda)}^{reg} = \overline{Z_r(\lambda)}^{reg}B(\lambda)^{-1}P^{-1} = \overline{U_r(\lambda)}^{reg}\overline{\Sigma_r(\lambda)}^{reg}\overline{V_r(\lambda)}^{reg\mathrm{T}}B(\lambda)^{-1}P^{-1}$.

Beside characterizing the global optimum of problem (7), we can also study the regularization path and relate it with the global optimum in the constrained function space. The following theorem states that the regularization path is continuous, and it connects the global optimum in the constrained function space and the global optimum without constraints or regularization. Although the regularization path for $\ell_2$ regularization is usually continuous in a vector space, in the case of rank constraints that we consider here the theorem is not trivial.

**Theorem 2.** *Assume* $\overline{Z(\lambda)}^{reg} = ZB(\lambda)^{-1}$ *is full rank for all* $\lambda \geq 0$. *Then, the regularization path of* $\widehat{W(\lambda)}^{reg}$ *is continuous on* $(0, \infty)$. *Moreover, we have* $\lim_{\lambda \to \infty} \widehat{W}^{reg}(\lambda) = \widehat{W}^{inv}$.

**Remark 3.** *Similar to Remark 2, the assumption that* $\overline{Z(\lambda)}^{reg}$ *is full rank for all* $\lambda \geq 0$ *is mild. If we fix any full row rank data matrix* $X$, *then* $B(\lambda)$ *is full rank for all* $\lambda \geq 0$. *Then, with probability* 1, $\overline{Z(\lambda)}^{reg} = ZB(\lambda)^{-1}$ *is full rank for all* $\lambda \geq 0$.

## 3.3 GLOBAL OPTIMUM IN FUNCTION SPACE WITH DATA AUGMENTATION

Data augmentation is another, data-driven, method to achieve invariance. As an informed regularization strategy, it increases the sample size by applying all possible group actions to the original data. The corresponding optimization problem is then given as follows:

$$\widehat{W} = \underset{W \in \mathbb{R}^{d_L \times d_0}}{\arg\min} \frac{1}{n|\mathcal{G}|} \sum_{g \in \mathcal{G}} \|W\rho_{\mathcal{X}}(g)X - Y\|_F^2, \quad \text{s.t.} \quad \mathrm{rank}(W) \leq r. \tag{9}$$

We can rewrite the above optimization problem in the following form:

$$\widehat{\widetilde{W}} = \underset{\widetilde{W} \in \mathbb{R}^{d_L \times d_0}}{\arg \min} \frac{1}{n|\mathcal{G}|} \|\widetilde{W} - |\mathcal{G}| Y X^{\mathrm{T}} \overline{G}^{\mathrm{T}} Q^{-1}\|_F^2, \quad \text{s.t. } \mathrm{rank}(\widetilde{W}) \leq r, \tag{10}$$

where $\overline{G} = \frac{1}{|\mathcal{G}|} \sum_{g \in \mathcal{G}} \rho_{\mathcal{X}}(g)$, and $Q$ is the square root of the symmetric positive definite matrix $\sum_{g \in \mathcal{G}} \rho_{\mathcal{X}}(g) X X^{\mathrm{T}} \rho_{\mathcal{X}}(g)^{\mathrm{T}}$, i.e., $Q^2 = \sum_{g \in \mathcal{G}} \rho_{\mathcal{X}}(g) X X^{\mathrm{T}} \rho_{\mathcal{X}}(g)^{\mathrm{T}}$. The following proposition characterizes the solution to the above optimization problem.

**Proposition 3.** *Denote $\overline{Z}^{da} = |\mathcal{G}| Y X^{\mathrm{T}} \overline{G}^{\mathrm{T}} Q^{-1}$, and $\overline{Z}^{da} = \overline{U}^{da} \overline{\Sigma}^{da} \overline{V}^{da}{}^{\mathrm{T}}$ as the SVD of $\overline{Z}^{da}$. Then the solution to the above optimization problem (9) is $\widehat{W}^{da} = \overline{Z}_r^{da} Q^{-1} = \overline{U}_r^{da} \overline{\Sigma}_r^{da} \overline{V}_r^{da}{}^{\mathrm{T}} Q^{-1}$. Moreover, if $\rho_{\mathcal{X}}$ is unitary, then $\widehat{W}^{da}$ is an invariant linear map, i.e., $\widehat{W}^{da} G = 0$.*

All together, we arrive at the following statement.

**Theorem 3.** *Assume $\rho_{\mathcal{X}}$ is unitary. Then the global optima in the function space with data augmentation and the global optima in the constrained function space are the same, i.e., $\widehat{W}^{da} = \widehat{W}^{inv}$.*

This theorem tells us that data augmentation and constrained model have the same global optima, which is also the limit of the global optima in the optimization problem with explicit regularization. Beside the global optima, we are also interested in comparing the critical points of the three optimization problems. The following section discusses this in detail.

### 3.4 CRITICAL POINTS IN THE FUNCTION SPACE

We consider a fixed matrix $Z \in \mathbb{R}^{d_L \times d_0}$ with SVD $Z = U \Sigma V^{\mathrm{T}}$. Let $m = \min\{d_0, d_L\}$ and denote by $[m]_r$ the set of all subsets of $[m]$ of cardinality $r$. For $\mathcal{I} \in [m]_r$, we define $\Sigma_{\mathcal{I}} \in \mathbb{R}^{d_L \times d_0}$ to be the diagonal matrix with entries $\sigma_{\mathcal{I},1}, \sigma_{\mathcal{I},2}, \ldots, \sigma_{\mathcal{I},m}$, where $\sigma_{\mathcal{I},i} = \sigma_i$ if $i \in \mathcal{I}$ and $\sigma_{\mathcal{I},i} = 0$ otherwise. Define $\ell_Z(W) := \|Z - W\|_F^2$ as the loss function in the function space $\mathcal{M}_r$. The function space $\mathcal{M}_r$ is a manifold with singularities. A point $P \in \mathcal{M}_r$ is a critical point of $\ell_Z$ if and only if $Z - P \in N_P \mathcal{M}_r$. Following Trager et al. (2020, Theorem 28) we can characterize the critical points of the loss function $\ell_Z$ in the function space $\mathcal{M}_r$ as follows (see Appendix A.8).

**Proposition 4.** *Assume all non-zero singular values of $\overline{Z}^{inv}, \overline{Z}^{da}, \overline{Z(\lambda)}^{reg}$ are pairwise distinct.*

1. *(Constrained Space) The number of critical points in the optimization problem (4) is $\binom{d}{r}$. They are all in the form of $\overline{U}^{inv} \overline{\Sigma}_{\mathcal{I}}^{inv} \overline{V}^{inv}{}^{\mathrm{T}} P^{-1}$, where $\mathcal{I} \in [d]_r$. The unique global minimum is $\overline{U}^{inv} \overline{\Sigma}_{[r]}^{inv} \overline{V}^{inv}{}^{\mathrm{T}} P^{-1}$, which is also the unique local minimum.*

2. *(Data Augmentation) The number of critical points in the optimization problem (9) is $\binom{d}{r}$. They are all in the form of $\overline{U}^{da} \overline{\Sigma}_{\mathcal{I}}^{da} \overline{V}^{da}{}^{\mathrm{T}} Q^{-1}$, where $\mathcal{I} \in [d]_r$. These critical points are the same as the critical points in the constrained function space. The unique global minimum is $\overline{U}^{da} \overline{\Sigma}_{[r]}^{da} \overline{V}^{da}{}^{\mathrm{T}} Q^{-1}$, which is also the unique local minimum.*

3. *(Regularization) The number of critical points in the optimization problem (7) is $\binom{m}{r}$. They are all in the form of $\overline{U}^{reg} \overline{\Sigma}_{\mathcal{I}}^{reg} \overline{V}^{reg}{}^{\mathrm{T}} B(\lambda)^{-1} P^{-1}$, where $\mathcal{I} \in [m]_r$. The unique global minimum is $\overline{U}^{reg} \overline{\Sigma}_{[r]}^{reg} \overline{V}^{reg}{}^{\mathrm{T}} B(\lambda)^{-1} P^{-1}$, which is also the unique local minimum.*

According to this result, we can say that the critical points in the constrained function space are the same as the critical points in the function space with data augmentation. Furthermore, the number of critical points in function space for a model trained with regularization is larger than the number of critical points in the other two cases.

We observe that fully-connected linear networks have no spurious local minima, meaning that each local minimum in parameter space corresponds to a local minimum in function space (Trager et al., 2020). This is a consequence of the geometry of determinantal varieties that also holds in our cases, suggesting that also for our three optimization problems there are no spurious local minima.

## 4 EXPERIMENTS

### 4.1 CONVERGENCE TO AN INVARIANT CRITICAL POINT VIA DATA AUGMENTATION

The following experiment demonstrates that gradient descent on the optimization problem (9) converges to a critical point that parameterizes an invariant function. The training data, consisting of 1000 samples before data augmentation, is a subset of the MNIST dataset. For computational efficiency, the images are downsampled to $14 \times 14$ pixels, resulting in a vectorized representation of dimension 196 for each image. The classification task involves 9 classes, and we aim to train a linear model mapping from $\mathbb{R}^{196}$ to $\mathbb{R}^9$ that is invariant under 90-degree rotations. Since digits 6 and 9 are rotationally equivalent, we exclude digit 9 from the dataset. The group associated with this invariance is the cyclic group of order 4, denoted as $\mathcal{G} = C_4$, where the representation $\rho_{\mathcal{X}}$ of $\mathcal{G}$ on $\mathbb{R}^{196}$ is the rotation operator. We employ a data augmentation technique that applies all possible group actions to the original data, yielding a total of 4000 training samples.

The model is a two-layer linear neural network with 5 hidden units, parameterizing all $\mathbb{R}^{9 \times 196}$ matrices with rank at most 5. We evaluate both *mean squared error (MSE)* and *cross-entropy (CE)* as the loss functions. For MSE, the targets are the one-hot encoded labels. The model is trained using the *Adam* optimizer (Kingma & Ba, 2015) with a learning rate of 0.001 and Adam parameter $\boldsymbol{\beta} = (0.9, 0.999)$, which is the default value in PyTorch (Paszke et al., 2019). The following Figure 1 depicts the evolution of certain entries in the end-to-end matrix $W$. In our setup, the learned linear map is invariant if and only if specific columns are identical. For example, according to the linear constraints in $W$ (see Proposition 1), columns 45, 52, 143, and 150 of $W$ should be exactly the same to achieve invariance. Figure 1a presents the results when trained with MSE, while Figure 1b shows the results with CE. In both cases, the entries in $W$ converge to approximately the same values, indicating that the learned map is nearly invariant. Additionally, we observe that the model trained with MSE converges significantly faster than the one trained with CE.

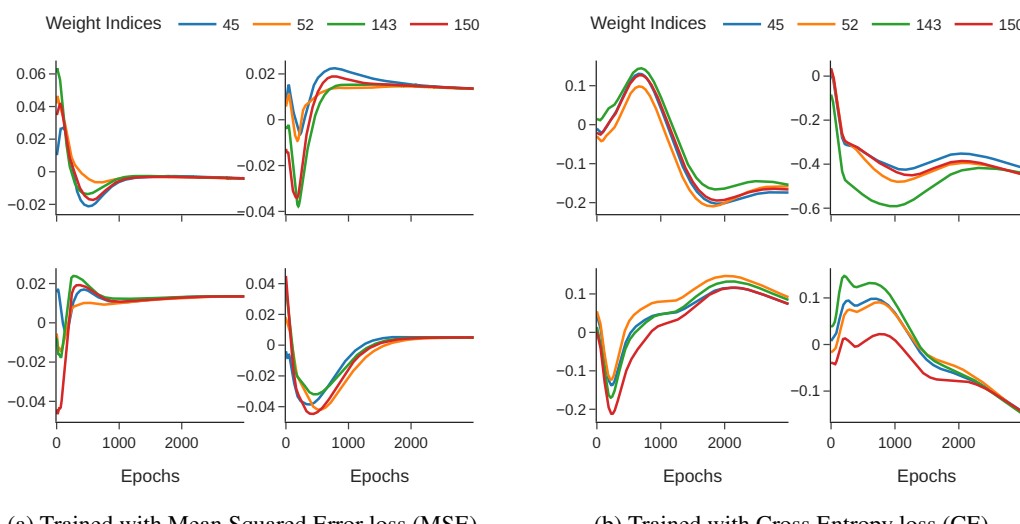

(a) Trained with Mean Squared Error loss (MSE).      (b) Trained with Cross Entropy loss (CE).

Figure 1: Weights in a two-layer linear neural network trained using Data Augmentation.

### 4.2 TRAINING CURVES OF ALL THREE APPROACHES

In the same setup as the previous experiment, we compare the performance of the model trained with all three approaches: data augmentation, hard-wiring, and regularization with different choices of the penalty parameter $\lambda$. In practice, we parameterize the model in the constrained function space by multiplying a basis matrix $B$ to the weight matrix of the linear model, i.e., $f(x) = W_2 W_1 B x$, where $W_2 \in \mathbb{R}^{9 \times 5}$ and $W_1 \in \mathbb{R}^{5 \times 49}$ are the learnable weight matrices of the linear model, and the basis matrix $B \in \mathbb{R}^{49 \times 196}$ is a matrix that satisfies $BG = 0$. It is worth noting that it is actually equivalent to perform feature-averaging before feeding the data to the model if we parameterize the

invariant function space in this way. Regarding the regularization method, $\lambda \in \{0.001, 0.01, 0.1\}$ when using MSE as the loss, and $\lambda \in \{0.01, 0.1, 1\}$ when using CE as the loss. We used the same data and setup as in the previous experiment.

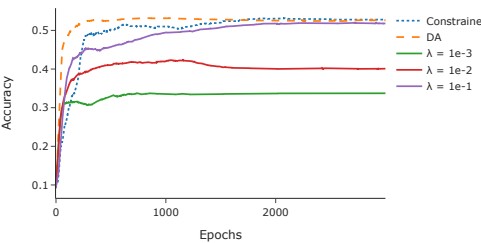

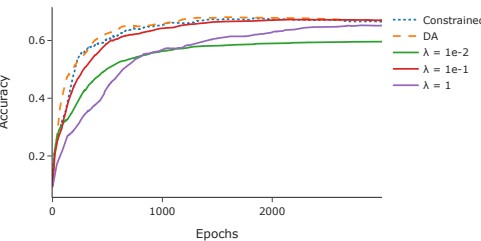

(a) Trained with Mean Squared Error loss (MSE).

(b) Trained with Cross Entropy loss (CE).

Figure 2: Training curves for Data Augmentation (DA), Regularization ($\lambda$), and Constrained model.

Figure 2 shows the training curves of all three methods under different losses. In terms of regularization, though the models are trained without data augmentation, the curves we show here are accuracy for the augmented dataset. We can see that data augmentation and hard-wiring have similar performance in the late stage of training. When $\lambda$ is suitable, regularization can also achieve very similar performance to the previous two methods. All three methods converge to the critical point at a similar rate (around 500 epochs). In fact, when trained with MSE, the hard-wired model converges to the same global optimum as the model trained with data augmentation. This result is consistent with the theoretical analysis in Theorem 3. Interestingly, even when trained with CE, all three methods have similar terminal performance. More experiments are needed to further investigate this phenomenon.

Regarding the amount of time required for training, training with data augmentation is computationally much more expensive than hard-wiring. This is because the model trained with data augmentation requires more samples (4 times more in this case) and more parameters (about 4 times more in this case) than the hard-wired model. Regularization is in between of the other two methods since it only requires more parameters but not more samples.

### 4.3 COMPARISON BETWEEN DATA AUGMENTATION AND REGULARIZATION

In this section, we empirically study the training dynamics in both data augmentation and regularization. Using the same setup as the above experiments, we are showing the evolution of the non-invariant part of the learned end-to-end matrix $\widehat{W}$. For any $\widehat{W}$, we can decompose it into two parts, an invariant part and a non-invariant part, i.e., $\widehat{W} = (\widehat{W} - \widehat{W}_\perp) + \widehat{W}_\perp$. A similar decomposition has been used by Gideoni (2023). In Figure 3, we track the evolution of $\widehat{W}_\perp$ by computing $\|\widehat{W}_\perp\|_F$ and $\|\widehat{W} - \widehat{W}_\perp\|_F^2 / \|\widehat{W}\|_F^2$ after each training epoch. When $\widehat{W}$ is very close to an invariant function, $\|\widehat{W}_\perp\|_F$ should be close to 0 and $\|\widehat{W} - \widehat{W}_\perp\|_F^2 / \|\widehat{W}\|_F^2$ should be close to 1. Figure 3 shows that with data augmentation $\|\widehat{W}_\perp\|_F$ first increases and then tends to decrease to zero. For regularization, since the penalty coefficient $\lambda$ is finite, the critical points are actually not invariant. Therefore, in this case we can see that $\|\widehat{W}_\perp\|_F$ does not converge to zero. Interestingly, we can see that for both data augmentation and regularization, $\|\widehat{W}_\perp\|_F$ displays a "double descent". Our conjecture is that the loss may also be decomposed into two parts, one controlling the error from invariance, and the other one controlling the error from the target. Therefore, the gradient of the weights during training can be decomposed into two directions as well, resulting in this phenomenon. This intuition could help us better understand the training dynamics and identify methods to accelerate training. Further research needs to be done to investigate this both theoretically and empirically. Experiments using cross entropy loss are included in Appendix A.10.

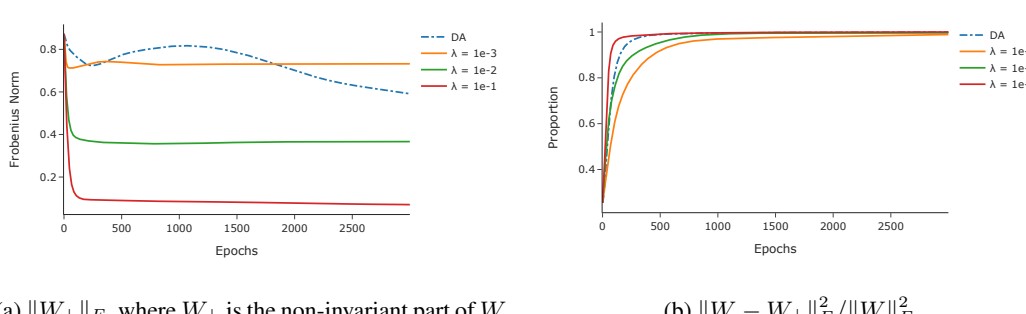

(a) $\|W_\perp\|_F$, where $W_\perp$ is the non-invariant part of $W$.      (b) $\|W - W_\perp\|_F^2 / \|W\|_F^2$.

Figure 3: Frobenius norm of the non-invariant part of the end-to-end matrix $W$, trained with Data Augmentation (DA) and Regularization ($\lambda$), with the Mean Squared Error loss (MSE).

## 5   CONCLUSION

This work explores learning with invariances from the perspective of the associated optimization problems. We investigate the loss landscape of linear invariant neural networks across the settings of data augmentation, constrained models, and explicit regularization, for which we characterized the form of the global optima (Proposition 3, Theorem 1, Proposition 2). We find that data augmentation and constrained models share the same global optima (Theorem 3), which also correspond to the limit of the global optima in the regularized problem (Theorem 2). Additionally, the critical points in both data augmentation and constrained models are identical, while regularization generally introduces more critical points (Proposition 4). Though our theoretical results are for linear networks, since we consider a non-convex function space, it is natural to conjecture that some of the conclusions might carry over to other models with non-convex function space. Empirical results in Appendix A.11 indicate that data augmentation and a constrained model indeed achieve a similar loss in the late phase of training for two-layer neural networks with different activation functions. At the same time, we observe that for models with a higher expressive power it is more difficult to learn invariance from the data. Based on our theoretical results, we suggest that data augmentation may have similar performance to constrained models, but will incur higher data and computing costs. The regularized model does not require more data and should have a performance close to the constrained model, but it may induce more critical points. The constrained model should have the best performance though one might need to design the invariant architecture carefully before feeding the data to the model.

**Limitations and future work**   We are focusing on deep linear networks, which are a simplified model of neural networks. Nonetheless, we considered the interesting case of rank-bounded end-to-end maps, which is a non-convex function space. Owing to the geometry of this model and the mean squared error loss (MSE), the global optima in all three optimization problems are the same. However, this is generally not true when the function class is more complicated or the loss is not the MSE. Moskalev et al. (2023) empirically suggest that data-driven methods fail to learn genuine invariance in weight-tying shallow ReLU networks for classification tasks with the cross-entropy loss. Experiments suggest that nonlinear networks may still learn invariance via data augmentation when trained with enough data. It would be interesting to investigate this phenomenon theoretically. Furthermore, as mentioned in Section 4.3, the training dynamics of our setup is also worth studying.

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

# A  APPENDIX

## A.1  PROOF OF PROPOSITION 1

**Proposition 1.** *Given a cyclic group $\mathcal{G}$ and a representation $\rho_{\mathcal{X}}$ of $\mathcal{G}$ on vector space $\mathcal{X} = \mathbb{R}^{d_0}$, a linear function $W$ mapping from $\mathcal{X}$ to $\mathcal{Y} = \mathbb{R}^{d_L}$ is invariant with respect to $\rho_{\mathcal{X}}$ if and only if $WG = 0$, where $G = \mathbf{I}_{d_0} - \rho_{\mathcal{X}}(g)$, and $g$ is the generator of $\mathcal{G}$.*

*Proof.* Suppose $\mathcal{G}$ is a cyclic group of order $k$ with generator $g$, i.e., $\mathcal{G} = \langle g \rangle$, $g^k = e$. If $W$ is invariant with respect to $\rho_{\mathcal{X}}$, then $W\rho_{\mathcal{X}}(h) = W$ for all $h \in \mathcal{G}$. Then we have $W(\mathbf{I}_{d_0} - \rho_{\mathcal{X}}(g)) = 0$ for the generator $g$.

Conversely, if $W(\mathbf{I}_{d_0} - \rho_{\mathcal{X}}(g)) = 0$ for the generator $g$, then we have $W\rho_{\mathcal{X}}(g) = W$. Multiplying both sides by $\rho_{\mathcal{X}}(g)$, we have $W\rho_{\mathcal{X}}(g^2) = W\rho_{\mathcal{X}}^2(g) = W\rho_{\mathcal{X}}(g) = W$. By induction, we can see that $W\rho_{\mathcal{X}}(g^j) = W$ for all $j \in [k]$. □

The following proposition extends the above proposition to cases when the group is continuous. The key point is that we can parameterize any element in the continuous group in terms of basis in its corresponding Lie algebra, along with a discrete set of generators.

**Proposition 5.** *[Theorem 1 in Finzi et al. (2021)] Let $\mathcal{G}$ be a real connected Lie group of dimension $M$ with finitely many connected components. Given a representation $\rho$ on vector space $V$ of dimension $D$, the constraint equations*

$$\rho(g)v = v, \forall v \in V, g \in \mathcal{G} \tag{11}$$

*holds if and only if*

$$d\rho(A_m)v = 0, \quad \forall m \in [M], \tag{12}$$
$$(\rho(h_p) - \mathbf{I}_D)v = 0, \quad \forall p \in [P], \tag{13}$$

*where $\{A_m\}_{m=1}^{M}$ are $M$ basis vectors for the $M$ dimensional Lie Algebra $\mathfrak{g}$ with induced representation $d\rho$, and for some finite collection $\{h_p\}_{p=1}^{P}$ of discrete generators.*

## A.2  EXTENSION FROM INVARIANCE TO EQUIVARIANCE

Extension from invariance to equivariance is straightforward due to the fact that the constraints are still linear in the vector space of linear maps from $\mathcal{X}$ to $\mathcal{Y}$. The following proposition shows how to find the linear constraints.

**Proposition 6.** *Given a group $\mathcal{G}$, an input vector space $\mathcal{X}$ with representation $\rho_{\mathcal{X}}$ of $\mathcal{G}$ and an output space $\mathcal{Y}$ with representation $\rho_{\mathcal{Y}}$ of $\mathcal{G}$, a linear function $f : \mathcal{X} \to \mathcal{Y}, x \mapsto Wx$ is equivariant with respect to $\rho_{\mathcal{X}}$ and $\rho_{\mathcal{Y}}$ if and only if $\mathrm{vec}(W) \in \bigcap_{g \in \mathcal{G}} \ker \left( \rho_{\mathcal{X}}(g) \otimes \rho_{\mathcal{Y}}(g^{-1})^{\mathrm{T}} - \mathbf{I}_{d_{\mathcal{X}} d_{\mathcal{Y}}} \right)$, where $d_{\mathcal{X}}$ is the dimension of $\mathcal{X}$ and $d_{\mathcal{Y}}$ is the dimension of $\mathcal{Y}$.*

*Proof.* By definition, $f$ is equivariant if and only if $W\rho_{\mathcal{X}}(g) = \rho_{\mathcal{Y}}(g)W$ for all $g \in \mathcal{G}$. We can then get $\rho_{\mathcal{Y}}(g^{-1})W\rho_{\mathcal{X}}(g) = W$. By vectorizing both sides, we can see that

$$\mathrm{vec}(\rho_{\mathcal{Y}}(g^{-1})W\rho_{\mathcal{X}}(g)) = \left( \rho_{\mathcal{X}}(g)^{\mathrm{T}} \otimes \rho_{\mathcal{Y}}(g^{-1}) \right) \mathrm{vec}(W) = \mathrm{vec}(W),$$

implying that $\mathrm{vec}(W) \in \bigcap_{g \in \mathcal{G}} \ker \left( \rho_{\mathcal{X}}(g) \otimes \rho_{\mathcal{Y}}(g^{-1})^{\mathrm{T}} - \mathbf{I}_{d_{\mathcal{X}} d_{\mathcal{Y}}} \right)$. □

### A.3 Proof of Theorem 1

The following lemma proves a key observation that if a matrix lives in a left null space of another matrix, then the low rank approximator remains in the left null space of the other matrix.

**Lemma 1.** *Given a matrix $A \in \mathbb{R}^{n \times m}$ and a matrix $B \in \mathbb{R}^{m \times p}$, $AB = 0$, where $d = \text{nullity}(B)$. Let $A = U\Sigma V^{\mathrm{T}}$ be the SVD of $A$, where $U \in \mathbb{R}^{n \times n}$, $\Sigma \in \mathbb{R}^{n \times m}$, and $V \in \mathbb{R}^{m \times m}$. Then for any $r \leq \text{rank}(A) \leq d$, $V_r^{\mathrm{T}}$ lives in the left null space of $B$, namely, $V_r^{\mathrm{T}} B = 0$, and $A_r B = 0$.*

*Proof.*

$$A = U\Sigma V^{\mathrm{T}}, \quad AB = 0$$
$$\Rightarrow \quad U\Sigma V^{\mathrm{T}} B = 0$$
$$\Rightarrow \quad \Sigma V^{\mathrm{T}} B = 0$$
$$\Rightarrow \quad \Sigma_d V_d^{\mathrm{T}} B = 0, \quad d = \text{nullity}(B).$$

Since $\Sigma_d$ is a diagonal matrix, and the diagonal entries are non-zero, we have that $V_d^{\mathrm{T}} B = 0$. And $V_d = [V_r \quad V_{d-r}]$, we have $V_r^{\mathrm{T}} B = 0$. We can now see that $A_r B = U_r \Sigma_r V_r^{\mathrm{T}} B = 0$. $\qquad\square$

**Theorem 1.** *Denote $\overline{Z}^{inv} := Z(\mathbf{I}_{d_0} - \widetilde{G}\widetilde{G}^+)$. We assume $\text{rank}(\overline{Z}^{inv}) > r$. Let $\overline{Z}^{inv} = \overline{U}^{inv}\overline{\Sigma}^{inv}\overline{V}^{inv\mathrm{T}}$ be the SVD of $\overline{Z}^{inv}$. Then the solution to (4) is $\widehat{W}^{inv} = \overline{U}_r^{inv}\overline{\Sigma}_r^{inv}\overline{V}_r^{inv\mathrm{T}} P^{-1}$.*

*Proof.* As stated in the main text, we can rewrite the optimization problem 4 as the following form:

$$\widehat{\widetilde{W}} = \underset{\widetilde{W} \in \mathbb{R}^{d_L \times d_0}}{\arg\min} \frac{1}{n}\|\widetilde{W} - Z\|_F^2, \quad \text{s.t.} \quad \widetilde{W}\widetilde{G} = 0, \ \text{rank}(\widetilde{W}) \leq r, \tag{14}$$

where $Z = YX^{\mathrm{T}}P^{-1}$, and $\widetilde{G} = P^{-1}G$. There are two cases to consider.

**Case 1**: $Z\widetilde{G} = 0$. We assume $Z$ has rank $d$. Then we can perform SVD on $Z = U\Sigma V^{\mathrm{T}} = U_d\Sigma_d V_d^{\mathrm{T}}$. Eckart & Young (1936b) have shown that the best rank-$r$ approximation of $Z$ is given by $Z_r = U_r\Sigma_r V_r^{\mathrm{T}}$. According to Lemma Lemma 1, we can see that $Z_r\widetilde{G} = 0$. Therefore, the solution to the above optimization problem is $\widehat{\widetilde{W}} = Z_r$.

**Case 2**: $Z\widetilde{G} \neq 0$. We can then decompose $Z = \overline{Z} + Z_\perp$, where $\overline{Z}\widetilde{G} = 0$, $\langle\overline{Z}, Z_\perp\rangle_F = 0$. Therefore, we can see that

$$\|\widetilde{W} - Z\|_F^2 = \|(\widetilde{W} - \overline{Z}) - Z_\perp\|_F^2$$
$$= \|\widetilde{W} - \overline{Z}\|_F^2 + \|Z_\perp\|_F^2 - 2\langle\widetilde{W} - \overline{Z}, Z_\perp\rangle_F$$
$$= \|\widetilde{W} - \overline{Z}\|_F^2 + \|Z_\perp\|_F^2 \tag{15}$$

Thus, the solution to the above optimization problem is

$$\widehat{\widetilde{W}} = \underset{\widetilde{W} \in \mathbb{R}^{d_L \times d_0}}{\arg\min} \|\widetilde{W} - Z\|_F^2 = \underset{\widetilde{W} \in \mathbb{R}^{d_L \times d_0}}{\arg\min} \|\widetilde{W} - \overline{Z}\|_F^2.$$

This is then reduced to the low-rank approximation problem of $\overline{Z}$, which is the same as in **Case 1**. Let $\overline{Z} = \overline{U}\overline{\Sigma}\overline{V}^{\mathrm{T}}$ be the SVD of $\overline{Z}$. Then the solution is $\widehat{\widetilde{W}} = \overline{U}_r\overline{\Sigma}_r\overline{V}_r^{\mathrm{T}}$.

Note that $\overline{Z}$ can be found by projecting $Z$ onto the left null space of $\widetilde{G}$. An easy construction is $\overline{Z} = Z(\mathbf{I}_{d_0} - \widetilde{G}\widetilde{G}^+)$. To see this, we can check that $\overline{Z}\widetilde{G} = 0$ and $\langle\overline{Z}, Z_\perp\rangle_F = 0$. We have

$$\overline{Z}\widetilde{G} = Z(\mathbf{I}_{d_0} - \widetilde{G}\widetilde{G}^+)\widetilde{G} = Z\widetilde{G} - Z\widetilde{G}\widetilde{G}^+\widetilde{G} = Z\widetilde{G} - Z\widetilde{G} = 0. \tag{16}$$

To check $\langle\overline{Z}, Z_\perp\rangle_F = 0$, we have

$$\langle\overline{Z}, Z_\perp\rangle_F = \text{tr}\left[\overline{Z}^{\mathrm{T}} Z_\perp\right] = \text{tr}\left[\overline{Z}^{\mathrm{T}}(Z - \overline{Z})\right]$$

$$= \mathrm{tr}\left[(\mathbf{I}_{d_0} - \widetilde{G}\widetilde{G}^+)^\mathrm{T} Z^\mathrm{T} Z \widetilde{G}\widetilde{G}^+\right]$$

$$= \mathrm{tr}\left[Z^\mathrm{T} Z \widetilde{G}\widetilde{G}^+\right] - \mathrm{tr}\left[\widetilde{G}\widetilde{G}^+ Z^\mathrm{T} Z \widetilde{G}\widetilde{G}^+\right]$$

$$= \mathrm{tr}\left[Z^\mathrm{T} Z \widetilde{G}\widetilde{G}^+\right] - \mathrm{tr}\left[Z^\mathrm{T} Z \widetilde{G}\widetilde{G}^+ \widetilde{G}\widetilde{G}^+\right]$$

$$= \mathrm{tr}\left[Z^\mathrm{T} Z \widetilde{G}\widetilde{G}^+\right] - \mathrm{tr}\left[Z^\mathrm{T} Z \widetilde{G}\widetilde{G}^+\right] = 0. \tag{17}$$

$\square$

## A.4 PROOF OF PROPOSITION 2

**Proposition 2.** *Denote* $B(\lambda)$ *the square root of the symmetric positive definite matrix* $\mathbf{I}_{d_0} + n\lambda\widetilde{G}\widetilde{G}^\mathrm{T}$, *i.e.,* $B(\lambda)^2 = \mathbf{I}_{d_0} + n\lambda\widetilde{G}\widetilde{G}^\mathrm{T}$. *Denote* $\overline{Z(\lambda)}^{reg} = ZB(\lambda)^{-1}$, *and* $\overline{Z(\lambda)}^{reg} = \overline{U(\lambda)}^{reg}\overline{\Sigma(\lambda)}^{reg}\overline{V(\lambda)}^{reg\mathrm{T}}$ *as the SVD of* $\overline{Z(\lambda)}^{reg}$. *Then the solution to problem 7 is* $\widehat{W(\lambda)}^{reg} = \overline{Z_r(\lambda)}^{reg}B(\lambda)^{-1}P^{-1} = \overline{U_r(\lambda)}^{reg}\overline{\Sigma_r(\lambda)}^{reg}\overline{V_r(\lambda)}^{reg\mathrm{T}}B(\lambda)^{-1}P^{-1}$.

*Proof.* The loss function is defined as:

$$\mathcal{L}(\widetilde{W}) = \frac{1}{n}\|\widetilde{W} - Z\|_F^2 + \lambda\|\widetilde{W}\widetilde{G}\|_F^2 \tag{18}$$

$$= \frac{1}{n}\mathrm{tr}[(\widetilde{W} - Z)^\mathrm{T}(\widetilde{W} - Z)] + \lambda\,\mathrm{tr}[(\widetilde{W}\widetilde{G})^\mathrm{T}(\widetilde{W}\widetilde{G})]$$

$$= \frac{1}{n}\mathrm{tr}[\widetilde{W}\widetilde{W}^\mathrm{T} - 2\widetilde{W}^\mathrm{T} Z + Z^\mathrm{T} Z] + \lambda\,\mathrm{tr}[\widetilde{W}(\widetilde{G}\widetilde{G}^\mathrm{T})\widetilde{W}^\mathrm{T}]$$

$$= \frac{1}{n}\mathrm{tr}[\widetilde{W}(\mathbf{I}_{d_0} + n\lambda\widetilde{G}\widetilde{G}^\mathrm{T})\widetilde{W}^\mathrm{T} - 2\widetilde{W}^\mathrm{T} Z + Z^\mathrm{T} Z]$$

$$= \frac{1}{n}\mathrm{tr}[\widetilde{W}B(\lambda)B(\lambda)^\mathrm{T}\widetilde{W}^\mathrm{T} - 2B(\lambda)^\mathrm{T}\widetilde{W}^\mathrm{T} ZB(\lambda)^{-1} + Z^\mathrm{T} Z]$$

$$= \frac{1}{n}\|\widetilde{W}B(\lambda) - ZB(\lambda)^{-1}\|_F^2 + \mathrm{const.} \tag{19}$$

Therefore, the optimization problem is equivalent to the following low rank approximation problem:

$$\widehat{\widetilde{W(\lambda)}} := \underset{\widetilde{W}\in\mathbb{R}^{d_L\times d_0}}{\arg\min} \frac{1}{n}\|\widetilde{W}B(\lambda) - ZB(\lambda)^{-1}\|_F^2, \quad \mathrm{rank}(\widetilde{W}) \le r \tag{20}$$

$$= \overline{Z_r(\lambda)}^{reg}B(\lambda)^{-1} \tag{21}$$

$$= \overline{U_r(\lambda)}^{reg}\overline{\Sigma_r(\lambda)}^{reg}\overline{V_r(\lambda)}^{reg\mathrm{T}}B(\lambda)^{-1} \tag{22}$$

Since $\widehat{W} = \widehat{\widetilde{W}}P^{-1}$, we have $\widehat{W(\lambda)}^{reg} = \overline{U_r(\lambda)}^{reg}\overline{\Sigma_r(\lambda)}^{reg}\overline{V_r(\lambda)}^{reg\mathrm{T}}B(\lambda)^{-1}P^{-1}$. $\square$

## A.5 PROOF OF THEOREM 2

To prove the theorem, we need the following lemma:

**Lemma 2** (Theorem 2.1 in Dieci et al. (2005)). *Let* $A$ *be a* $\mathcal{C}^s, s \ge 1$, *matrix valued function,* $t \in [0,1] \to A(t) \in \mathbb{R}^{m\times n}, m \ge n$, *of rank* $n$, *having* $p$ *disjoint groups of singular values (* $p \le n$ *) that vary continuously for all* $t$ *:* $\Sigma_1, \dots, \Sigma_p$. *Let* $z = m - n$. *Consider the function* $M \in \mathcal{C}^s\left([0,1], \mathbb{R}^{(m+n)\times(m+n)}\right)$ *given by*

$$M(t) = \begin{bmatrix} 0 & A(t) \\ A^\mathrm{T}(t) & 0 \end{bmatrix}. \tag{23}$$

*Then, there exists orthogonal* $Q \in \mathcal{C}^s\left([0,1], \mathbb{R}^{(m+n)\times(m+n)}\right)$ *of the form*

$$Q(t) = \begin{bmatrix} U_2(t) & U_1(t)/\sqrt{2} & U_1(t)/\sqrt{2} \\ 0 & V(t)/\sqrt{2} & -V(t)/\sqrt{2} \end{bmatrix}, \tag{24}$$

*such that*

$$Q^{\mathrm{T}}(t)M(t)Q(t) = \begin{bmatrix} 0 & 0 & 0 \\ 0 & S(t) & 0 \\ 0 & 0 & -S(t) \end{bmatrix}, \tag{25}$$

*where $S$ is $S = \mathrm{diag}\,(S_i, i = 1,\ldots,p)$, and each $S_i$ is symmetric positive definite, and its eigenvalues coincide with the $\Sigma_i, i = 1,\ldots,p$. We have $U_2 \in \mathcal{C}^s\,([0,1],\mathbb{R}^{m\times z})\,, U_1 \in \mathcal{C}^s\,([0,1],\mathbb{R}^{m\times n}),$ and $V \in \mathcal{C}^s\,([0,1],\mathbb{R}^{n\times n})$. Equivalently, if we let $U = [\; U_1 \quad U_2 \;]$, then*

$$U^{\mathrm{T}}(t)A(t)V(t) = \begin{bmatrix} S(t) \\ 0 \end{bmatrix},$$

*with the previous form of $S$.*

**Theorem 2.** *Assume $\overline{Z(\lambda)}^{reg} = ZB(\lambda)^{-1}$ is full rank for all $\lambda \geq 0$. Then, the regularization path of $\widehat{W(\lambda)}^{reg}$ is continuous on $(0,\infty)$. Moreover, we have $\lim_{\lambda\to\infty} \widehat{W}^{reg}(\lambda) = \widehat{W}^{inv}$.*

*Proof.* Let $U^{\widetilde{G}}\Sigma^{\widetilde{G}}V^{\widetilde{G}^{\mathrm{T}}}$ be the SVD of $\widetilde{G}$. Since $\mathrm{nullity}(\widetilde{G}) = d$, then $\mathrm{rank}(\widetilde{G}) = d_0 - d$, suggesting that only the first $d_0 - d$ elements of $\Sigma^{\widetilde{G}}$ are non-zero. Denote $\Sigma^{\widetilde{G}} = \mathrm{diag}(\sigma_1^{\widetilde{G}},\ldots,\sigma_{d_0-d}^{\widetilde{G}},0,\ldots,0)$, then we have $\widetilde{G}^+ = V^{\widetilde{G}}\,\mathrm{diag}(1/\sigma_1^{\widetilde{G}},\ldots,1/\sigma_{d_0-d}^{\widetilde{G}},0,\ldots,0)U^{\widetilde{G}^{\mathrm{T}}}$ according to the property of Moore-Penrose pseudoinverse. Therefore, we have

$$\mathbf{I}_{d_0} + n\lambda\widetilde{G}\widetilde{G}^{\mathrm{T}} = \mathbf{I}_{d_0} + n\lambda U^{\widetilde{G}}\Sigma^{\widetilde{G}^2}U^{\widetilde{G}^{\mathrm{T}}} = U^{\widetilde{G}}\left(\mathbf{I}_{d_0} + n\lambda\Sigma^{\widetilde{G}^2}\right)U^{\widetilde{G}^{\mathrm{T}}}$$

$$= U^{\widetilde{G}}\,\mathrm{diag}(1 + n\lambda\sigma_1^{\widetilde{G}^2},\ldots,1 + n\lambda\sigma_{d_0-d}^{\widetilde{G}^2},1,\ldots,1)U^{\widetilde{G}^{\mathrm{T}}}, \tag{26}$$

$$B(\lambda) := (\mathbf{I}_{d_0} + n\lambda\widetilde{G}\widetilde{G}^{\mathrm{T}})^{\frac{1}{2}}$$

$$= U^{\widetilde{G}}\,\mathrm{diag}(\sqrt{1 + n\lambda\sigma_1^{\widetilde{G}^2}},\ldots,\sqrt{1 + n\lambda\sigma_{d_0-d}^{\widetilde{G}^2}},1,\ldots,1)U^{\widetilde{G}^{\mathrm{T}}}, \tag{27}$$

and

$$\lim_{\lambda\to\infty} B(\lambda)^{-1} = \lim_{\lambda\to\infty} (\mathbf{I}_{d_0} + n\lambda\widetilde{G}\widetilde{G}^{\mathrm{T}})^{-\frac{1}{2}}$$

$$= \lim_{\lambda\to\infty} U^{\widetilde{G}}\,\mathrm{diag}(1/\sqrt{1 + n\lambda\sigma_1^{\widetilde{G}^2}},\ldots,1/\sqrt{1 + n\lambda\sigma_{d_0-d}^{\widetilde{G}^2}},1,\ldots,1)U^{\widetilde{G}^{\mathrm{T}}},$$

$$= U^{\widetilde{G}}\,\mathrm{diag}(0,\ldots,0,1,\ldots,1)U^{\widetilde{G}^{\mathrm{T}}} \tag{28}$$

On the other hand, we have

$$\mathbf{I}_{d_0} - \widetilde{G}\widetilde{G}^+ = \mathbf{I}_{d_0} - U^{\widetilde{G}}\,\mathrm{diag}(1,\ldots,1,0,\ldots,0)U^{\widetilde{G}^{\mathrm{T}}}$$

$$= U^{\widetilde{G}}\,\mathrm{diag}(0,\ldots,0,1,\ldots,1)U^{\widetilde{G}^{\mathrm{T}}}.$$

Thus, we can see that $\lim_{\lambda\to\infty} B(\lambda)^{-1} = \mathbf{I}_{d_0} - \widetilde{G}\widetilde{G}^+$.

Recall that $\widehat{W(\lambda)}^{reg} = \overline{Z_r(\lambda)}^{reg}B(\lambda)^{-1}P^{-1} = \overline{U_r(\lambda)}^{reg}\overline{\Sigma_r(\lambda)}^{reg}\overline{V_r(\lambda)}^{reg\mathrm{T}}B(\lambda)^{-1}P^{-1}$ and $\widehat{W}^{inv} = \overline{U}_r^{inv}\overline{\Sigma}_r^{inv}\overline{V}_r^{inv\mathrm{T}}$.

First, we want to show that the regularization path is continuous on $(0,\infty)$. According to Weyl's inequality for singular values, we have the following inequalities:

$$|\sigma_k(\overline{Z(\lambda+\delta)}^{reg}) - \sigma_k(\overline{Z(\lambda)}^{reg})| \leq \|\overline{Z(\lambda+\delta)}^{reg} - \overline{Z(\lambda)}^{reg}\|_2, \quad \forall k \in [\min\{d_0, d_L\}]. \tag{29}$$

On the other hand, we have,

$$\|\overline{Z(\lambda+\delta)}^{reg} - \overline{Z(\lambda)}^{reg}\|_2 \tag{30}$$

$$= \|ZB(\lambda+\delta)^{-1} - ZB(\lambda)^{-1}\|_2 \tag{31}$$

$$= \|ZU^{\widetilde{G}} \operatorname{diag}\left(\frac{1}{\sqrt{1+n\lambda\sigma_1^{\widetilde{G}^2}}} - \frac{1}{\sqrt{1+n(\lambda+\delta)\sigma_1^{\widetilde{G}^2}}}, \ldots, \right. \tag{32}$$

$$\left. \frac{1}{\sqrt{1+n\lambda\sigma_{d_0-d}^{\widetilde{G}}}^2} - \frac{1}{\sqrt{1+n(\lambda+\delta)\sigma_{d_0-d}^{\widetilde{G}}}^2}, 0, \ldots, 0 \right) U^{\widetilde{G}^{\mathrm{T}}}\|_2 \tag{33}$$

$$\le \|Z\|_2 \max_{i\in[d_0-d]} \left| \frac{1}{\sqrt{1+n\lambda\sigma_i^{\widetilde{G}^2}}} - \frac{1}{\sqrt{1+n(\lambda+\delta)\sigma_i^{\widetilde{G}^2}}} \right| \to 0, \quad \text{as } \delta \to 0. \tag{34}$$

Therefore, the singular values of $\overline{Z(\lambda)}^{reg}$ are continuous with respect to $\lambda$ on $(0,\infty)$. It is also easy to check that the function $f(\lambda) = \frac{1}{\sqrt{1+c\lambda}}$ is smooth on $[0,\infty)$ for any constant $c > 0$. Applying Lemma 2 to $\overline{Z(\lambda)}^{reg}$, we find that there exist smooth $\overline{U(\lambda)}^{reg}$ and $\overline{V(\lambda)}^{reg}$ such that $\overline{Z(\lambda)}^{reg} = \overline{U(\lambda)}^{reg}\overline{\Sigma(\lambda)}^{reg}\overline{V(\lambda)}^{reg\mathrm{T}}$. Thus, by truncating $\overline{U(\lambda)}^{reg}$ and $\overline{V(\lambda)}^{reg}$, $\overline{U_r(\lambda)}^{reg}$ and $\overline{V_r(\lambda)}^{reg}$ are also smooth functions of $\lambda$ on $(0,\infty)$. Since the singular values are continuous with respect to $\lambda$, we have that $\overline{\Sigma_r(\lambda)}^{reg}$ is also continuous on $(0,\infty)$. Then $B(\lambda)$ is continuous on $(0,\infty)$. Since the product of continuous functions is continuous, the regularization path is continuous on $(0,\infty)$.

Finally, we want to show that $\lim_{\lambda\to\infty}\widehat{W(\lambda)}^{reg} = \widehat{W}^{inv}$. We notice that $\lim_{\lambda\to\infty}\overline{Z_r(\lambda)}^{reg} = \lim_{\lambda\to\infty} ZB(\lambda)^{-1} = Z(\mathbf{I}_{d_0} - \widetilde{G}\widetilde{G}^+) = \overline{Z}^{inv}$. According to the continuity of the regularization path, we get $\lim_{\lambda\to\infty}\overline{U_r(\lambda)}^{reg}\overline{\Sigma_r(\lambda)}^{reg}\overline{V_r(\lambda)}^{reg\mathrm{T}} = \overline{U}_r^{inv}\overline{\Sigma}_r^{inv}\overline{V}_r^{inv\mathrm{T}}$.

Due to the fact that $\lim_{\lambda\to\infty} ZB(\lambda)^{-1}$ lives in the left null space of $\widetilde{G}$, Lemma 1 tells us that $\lim_{\lambda\to\infty}\overline{U_r(\lambda)}^{reg}\overline{\Sigma_r(\lambda)}^{reg}\overline{V_r(\lambda)}^{reg\mathrm{T}}$ also lives in the left null space of $\widetilde{G}$. Thus, we have that

$$\lim_{\lambda\to\infty} \overline{U_r(\lambda)}^{reg}\overline{\Sigma_r(\lambda)}^{reg}\overline{V_r(\lambda)}^{reg\mathrm{T}} B(\lambda)^{-1} = \lim_{\lambda\to\infty} \overline{U_r(\lambda)}^{reg}\overline{\Sigma_r(\lambda)}^{reg}\overline{V_r(\lambda)}^{reg\mathrm{T}}. \tag{35}$$

The proof is complete. $\qquad\square$

## A.6 Proof of Proposition 3

To prove the proposition, we need the following lemma:

**Lemma 3.** *$M$ and $G$ are both real $d$ by $d$ matrices. $G$ is diagonalizable, and $M$ is positive definite. If $MG = GM$, then $M^{\frac{1}{2}}G = GM^{\frac{1}{2}}$, where $M^{\frac{1}{2}}$ is the positive definite square root of $M$.*

*Proof.* Let $M = P\Lambda P^{\mathrm{T}}$ be the eigen decomposition of $M$. Since $M$ is positive definite, we have that $P$ is orthogonal, and $\Lambda$ is a diagonal matrix with positive entries. According to theorem 1.3.12 in Horn & Johnson (2017), we know that $PGP^{\mathrm{T}}$ is also diagonal since $M$ and $G$ commute. Write $G = PDP^{\mathrm{T}}$, then $GM^{\frac{1}{2}} = P^{\mathrm{T}}DPP^{\mathrm{T}}\Lambda P = P^{\mathrm{T}}D\Lambda P = P^{\mathrm{T}}\Lambda PP^{\mathrm{T}}DP = M^{\frac{1}{2}}G$. $\qquad\square$

**Lemma 4.** *Let $(\mathcal{G}, \mathcal{A}, \lambda)$ be a measure space. Consider a nontrivial representation $\rho_{\mathcal{X}}$ of a compact group $\mathcal{G}$, let $\lambda$ be the normalized Haar measure on $\mathcal{G}$. The existence of the Haar measure is guaranteed by the compactness of $\mathcal{G}$ (Bourbaki, 2004). Define $\overline{G} := \int_{\mathcal{G}} \rho_{\mathcal{X}}(g)d\lambda(g)$. Then we have the following properties:*

1. *$\overline{G}\rho_{\mathcal{X}}(h) = \overline{G}$ for all $h \in \mathcal{G}$.*

2. *$\overline{G}$ is idempotent, i.e., $\overline{G}^2 = \overline{G}$. That is to say, $\overline{G}$ is a projection operator from $\mathcal{X}$ to the subspace all $\mathcal{G}$-fixed points.*

3. *If $\rho_{\mathcal{X}}$ is unitary, i.e., $\rho_{\mathcal{X}}(h)^{\dagger}\rho_{\mathcal{X}}(h) = \mathbf{I}_d$ for all $h \in \mathcal{G}$, then $\overline{G}$ is Hermitian.*

*Proof.*

1. Here, we need to use the fact that the Haar measure is left-invariant, i.e., $\lambda(gA) = \lambda(A)$ for all $g \in \mathcal{G}$ and $A \in \mathcal{A}$. We have

$$\overline{G}\rho_{\mathcal{X}}(h) = \int_{\mathcal{G}} \rho_{\mathcal{X}}(g)d\lambda(g)\rho_{\mathcal{X}}(h) = \int_{\mathcal{G}} \rho_{\mathcal{X}}(gh)d\lambda(g) = \int_{\mathcal{G}} \rho_{\mathcal{X}}(gh)d\lambda(gh) = \overline{G}. \quad (36)$$

2. To show that $\overline{G}$ is idempotent, we have

$$\overline{G}^2 = \left( \int_{\mathcal{G}} \rho_{\mathcal{X}}(g)d\lambda(g) \right) \left( \int_{\mathcal{G}} \rho_{\mathcal{X}}(h)d\lambda(h) \right) = \int_{\mathcal{G}} \int_{\mathcal{G}} \rho_{\mathcal{X}}(g)\rho_{\mathcal{X}}(h)d\lambda(g)d\lambda(h)$$

$$= \int_{\mathcal{G}} \int_{\mathcal{G}} \rho_{\mathcal{X}}(gh)d\lambda(g)d\lambda(h) = \int_{\mathcal{G}} \int_{\mathcal{G}} \rho_{\mathcal{X}}(gh)d\lambda(gh)d\lambda(h) = \int_{\mathcal{G}} \overline{G}d\lambda(h) = \overline{G}.$$

$$(37)$$

3. To see the last property, we have

$$\overline{G}^\dagger = \int_{\mathcal{G}} \rho_{\mathcal{X}}(g)^\dagger d\lambda(g) = \int_{\mathcal{G}} \rho_{\mathcal{X}}(g)^{-1}d\lambda(g) = \int_{\mathcal{G}} \rho_{\mathcal{X}}(g)d\lambda(g) = \overline{G}. \quad (38)$$

$\square$

**Lemma 5.** *Given a finite group $\mathcal{G}$ with order $n$ and a representation $\rho$ of $\mathcal{G}$ on vector space $V$ over field $\mathbb{C}$, then for every $g \in \mathcal{G}$, there exists a basis $P_g$ in which the matrix of $\rho(g)$ is diagonal for all $g \in \mathcal{G}$, with $n$-th roots of unity on the diagonal.*

*Proof.* Since $\mathcal{G}$ is finite with order $n$, let $g$ be the generator of $\mathcal{G}$, i.e., $g^n = e$ and $\rho(g)^n = \rho(g^n) = \rho(e) = \mathbf{I}$. We can write $\rho(g)$ in the form of Jordan canonical form, i.e., $\rho(g) = P_g^{-1}JP_g$, where $P_g \in GL(V)$, $J$ is a block diagonal matrix in the following form

$$J = \begin{bmatrix} J_1 & & \\ & \ddots & \\ & & J_p \end{bmatrix},$$

and each block $J_i$ is a square matrix of the form

$$J_i = \begin{bmatrix} \lambda_i & 1 & & \\ & \lambda_i & \ddots & \\ & & \ddots & 1 \\ & & & \lambda_i \end{bmatrix}.$$

We know that $\rho(g)^n = \mathbf{I}$, then $J^n = \mathbf{I}$, which implies that $J_i^n = \mathbf{I}$ for all $i \in [p]$. Let $N_i$ be the Jordan block matrix with $\lambda_i = 0$. Then

$$J_i^n = (\lambda_i\mathbf{I} + N_i)^n = \sum_{k=0}^{n} \binom{n}{k} \lambda_i^{n-k}N_i^k = \mathbf{I}.$$

Notice that $N_i^q$ is the matrix with zeros and ones only, with the ones in index position $(a, b)$ with $a = b + q$. Therefore, the sum can be $\mathbf{I}$ if and only if $\lambda_i^n = 1$ and $N_i = \mathbf{0}$ for all $i \in [p]$. Therefore, $\lambda_i$ is an $n$-th root of unity for all $i \in [p]$, and $J_i$ is diagonal with $n$-th roots of unity on the diagonal. Let $m \in [n]$, then $\rho(g^m) = \rho(g)^m = P_g^{-1}J^mP_g$. Clearly, $J^m$ is also a diagonal matrix with $n$-th roots of unity on the diagonal. Therefore, the basis $P_g$ is the same for all $\rho(g^m)$. $\square$

**Proposition 3.** *Denote $\overline{Z}^{da} = |\mathcal{G}|YX^{\mathrm{T}}\overline{G}^{\mathrm{T}}Q^{-1}$, and $\overline{Z}^{da} = \overline{U}^{da}\overline{\Sigma}^{da}\overline{V}^{da\,\mathrm{T}}$ as the SVD of $\overline{Z}^{da}$. Then the solution to the above optimization problem (9) is $\widehat{W}^{da} = \overline{Z}_r^{da}Q^{-1} = \overline{U}_r^{da}\overline{\Sigma}_r^{da}\overline{V}_r^{da\,\mathrm{T}}Q^{-1}$. Moreover, if $\rho_{\mathcal{X}}$ is unitary, then $\widehat{W}^{da}$ is an invariant linear map, i.e., $\widehat{W}^{da}G = 0$.*

*Proof.* It is easy to see that $\widehat{W}^{da} = \overline{U}_r^{da}\overline{\Sigma}_r^{da}\overline{V}_r^{da\,\mathrm{T}}Q^{-1}$ is the solution to the optimization problem 9 since it is in the exact form of a low-rank approximation, and we can apply the Eckart-Young-Mirsky theorem Eckart & Young (1936b) to get the solution directly. We still need to check that $\widehat{W}^{da}$ is an invariant linear map, i.e., $\widehat{W}^{da}G = 0$. We have First, we observe that $\left(\sum_{g \in \mathcal{G}} \rho_{\mathcal{X}}(g)XX^{\mathrm{T}}\rho_{\mathcal{X}}(g)^{\mathrm{T}}\right)^{-1}\rho_{\mathcal{X}}(h) = \rho_{\mathcal{X}}(h)\left(\sum_{g \in \mathcal{G}} \rho_{\mathcal{X}}(g)XX^{\mathrm{T}}\rho_{\mathcal{X}}(g)^{\mathrm{T}}\right)^{-1}$ for all $h \in \mathcal{G}$. To see this, we have

$$
\left(\sum_{g \in \mathcal{G}} \rho_{\mathcal{X}}(g)XX^{\mathrm{T}}\rho_{\mathcal{X}}(g)^{\mathrm{T}}\right)^{-1}\rho_{\mathcal{X}}(h)
$$

$$
= \left(\sum_{g \in \mathcal{G}} \rho_{\mathcal{X}}(h^{-1})\rho_{\mathcal{X}}(g)XX^{\mathrm{T}}\rho_{\mathcal{X}}(g)^{\mathrm{T}}\right)^{-1}
$$

$$
= \rho_{\mathcal{X}}(h^{-1})^{\mathrm{T}}\left(\sum_{g \in \mathcal{G}} \rho_{\mathcal{X}}(h^{-1})\rho_{\mathcal{X}}(g)XX^{\mathrm{T}}\rho_{\mathcal{X}}(g)^{\mathrm{T}}\rho_{\mathcal{X}}(h^{-1})^{\mathrm{T}}\right)^{-1} \quad \text{unitarity of } \rho_{\mathcal{X}}
$$

$$
= \rho_{\mathcal{X}}(h)\left(\sum_{g \in \mathcal{G}} \rho_{\mathcal{X}}(h^{-1}g)XX^{\mathrm{T}}\rho_{\mathcal{X}}(h^{-1}g)^{\mathrm{T}}\right)^{-1}. \tag{39}
$$

Then by Lemma 3, we have $Q^{-1}\rho_{\mathcal{X}}(h) = \rho_{\mathcal{X}}(h)Q^{-1}$. And, we have $\overline{G} = \overline{G}\rho_{\mathcal{X}}(h)$ for all $h \in \mathcal{G}$ by Lemma 4. Therefore, we have

$$
\overline{Z}^{da}\rho_{\mathcal{X}}(h) = |\mathcal{G}|YX^{\mathrm{T}}\overline{G}^{\mathrm{T}}Q^{-1}\rho_{\mathcal{X}}(h)
$$

$$
= |\mathcal{G}|YX^{\mathrm{T}}\overline{G}^{\mathrm{T}}\rho_{\mathcal{X}}(h)Q^{-1}
$$

$$
= |\mathcal{G}|YX^{\mathrm{T}}\overline{G}^{\mathrm{T}}Q^{-1} = \overline{Z}^{da}. \tag{40}
$$

Thus, we can say that $\overline{Z}^{da}G = 0$. Based on Lemma 1, we can get that $\overline{Z}_r^{da}G = \overline{U}_r^{da}\overline{\Sigma}_r^{da}\overline{V}_r^{da\,\mathrm{T}}G = 0$. Therefore,

$$
\widehat{W}^{da}\rho_{\mathcal{X}}(h) = \overline{U}_r^{da}\overline{\Sigma}_r^{da}\overline{V}_r^{da\,\mathrm{T}}Q^{-1}\rho_{\mathcal{X}}(h)
$$

$$
= \overline{U}_r^{da}\overline{\Sigma}_r^{da}\overline{V}_r^{da\,\mathrm{T}}\rho_{\mathcal{X}}(h)Q^{-1}
$$

$$
= \overline{U}_r^{da}\overline{\Sigma}_r^{da}\overline{V}_r^{da\,\mathrm{T}}Q^{-1} = \widehat{W}^{da}. \tag{41}
$$

$\square$

### A.7 PROOF OF THEOREM 3

To prove the theorem, we need the following lemma:

**Lemma 6.** *Let $A = \begin{bmatrix} A_{11} & A_{21}^{\dagger} \\ A_{21} & A_{22} \end{bmatrix} \in \mathrm{GL}(n + m, \mathbb{C})$ be Hermitian and positive definite and $B \in \mathrm{GL}(n, \mathbb{C})$, where $A_{11} \in \mathrm{GL}(n, \mathbb{C})$ and $A_{22} \in \mathrm{GL}(m, \mathbb{C})$ are both Hermitian and positive definite. Define $E = A \times \begin{bmatrix} B & 0_{n,m} \\ 0_{m,n} & 0_{m,m} \end{bmatrix} = \begin{bmatrix} A_{11}B & 0_{n,m} \\ A_{21}B & 0_{m,m} \end{bmatrix}$. Then $E^{+} = \begin{bmatrix} E_{11} & E_{12} \\ 0_{m,n} & 0_{m,m} \end{bmatrix}$, where $E_{11} = B^{-1}\left(A_{11}^2 + A_{21}^{\dagger}A_{21}\right)^{-1}A_{11}$, and $E_{12} = B^{-1}\left(A_{11}^2 + A_{21}^{\dagger}A_{21}\right)^{-1}A_{21}^{\dagger}$.*

*Proof.* We need to verify that our solution satisfies the properties of the Moore-Penrose pseudoinverse. Notice the following property:

$$
E_{11}A_{11} + E_{12}A_{21} = B^{-1} \tag{42}
$$

First, we need to show that $EE^+E = E$ and $E^+EE^+ = E^+$. We have

$$EE^+E = \begin{bmatrix} A_{11}B & 0_{n,m} \\ A_{21}B & 0_{m,m} \end{bmatrix} \begin{bmatrix} E_{11} & E_{12} \\ 0_{m,n} & 0_{m,m} \end{bmatrix} \begin{bmatrix} A_{11}B & 0_{n,m} \\ A_{21}B & 0_{m,m} \end{bmatrix}$$

$$= \begin{bmatrix} A_{11}BE_{11} & A_{11}BE_{12} \\ A_{21}BE_{11} & A_{21}BE_{12} \end{bmatrix} \begin{bmatrix} A_{11}B & 0_{n,m} \\ A_{21}B & 0_{m,m} \end{bmatrix}$$

$$= \begin{bmatrix} A_{11}B(E_{11}A_{11} + E_{12}A_{21})B & 0_{n,m} \\ A_{21}B(E_{11}A_{11} + E_{12}A_{21})B & 0_{m,m} \end{bmatrix}$$

$$= \begin{bmatrix} A_{11}B & 0_{n,m} \\ A_{21}B & 0_{m,m} \end{bmatrix} = E. \tag{43}$$

Similarly, we want to show that $E^+EE^+ = E^+$. We have

$$E^+EE^+ = \begin{bmatrix} E_{11} & E_{12} \\ 0_{m,n} & 0_{m,m} \end{bmatrix} \begin{bmatrix} A_{11}B & 0_{n,m} \\ A_{21}B & 0_{m,m} \end{bmatrix} \begin{bmatrix} E_{11} & E_{12} \\ 0_{m,n} & 0_{m,m} \end{bmatrix}$$

$$= \begin{bmatrix} (E_{11}A_{11} + E_{12}A_{21})B & 0_{n,m} \\ 0_{m,n} & 0_{m,m} \end{bmatrix} \begin{bmatrix} E_{11} & E_{12} \\ 0_{m,n} & 0_{m,m} \end{bmatrix}$$

$$= \begin{bmatrix} \mathbf{I}_n & 0_{n,m} \\ 0_{m,n} & 0_{m,m} \end{bmatrix} \begin{bmatrix} E_{11} & E_{12} \\ 0_{m,n} & 0_{m,m} \end{bmatrix}$$

$$= \begin{bmatrix} E_{11} & E_{12} \\ 0_{m,n} & 0_{m,m} \end{bmatrix} = E^+. \tag{44}$$

We also need to verify that $EE^+$ and $E^+E$ are Hermitian. We have

$$EE^+ = \begin{bmatrix} A_{11} \left(A_{11}^2 + A_{21}^\dagger A_{21}\right)^{-1} A_{11} & A_{11} \left(A_{11}^2 + A_{21}^\dagger A_{21}\right)^{-1} A_{21}^\dagger \\ A_{21} \left(A_{11}^2 + A_{21}^\dagger A_{21}\right)^{-1} A_{11} & A_{21} \left(A_{11}^2 + A_{21}^\dagger A_{21}\right)^{-1} A_{21}^\dagger \end{bmatrix}, \tag{45}$$

and

$$E^+E = \begin{bmatrix} \mathbf{I}_n & 0_{n,m} \\ 0_{m,n} & 0_{m,m} \end{bmatrix}. \tag{46}$$

It is clear that both $EE^+$ and $E^+E$ are Hermitian. Therefore, we have shown that $E^+$ is indeed the Moore-Penrose pseudoinverse of $E$. $\square$

**Lemma 7.** *Let $Z \in \mathbb{C}^{m \times n}$ be a full-rank matrix. $Q \in \mathbb{C}^{n \times n}$ is Hermitian and positive semi-definite, and $P \in \mathbb{C}^{n \times n}$ satisfying $Q^2 = PP^\dagger$. Given $r < \text{rank}(Q)$, let $Z_1$ and $Z_2$ be the best rank-$r$ approximation of $ZQ$ and $ZP$ with respect to the Frobenius norm, respectively, then $Z_1Q = Z_2P^\dagger$.*

*Proof.* Let $P = USV^\dagger$ be the SVD of $P$, then we have $Q = USU^\dagger$. Since $ZQ^2 = ZPP^\dagger$, we can see that $ZQUSU^\dagger = ZPVSU^\dagger$. Therefore, we have $ZQ = ZP(VU^\dagger)$. $VU^\dagger$ is a unitary matrix, and according to the rotational invariance of SVD, we can say that $Z_1 = Z_2(VU^\dagger)$, i.e., if $ZP = \widetilde{U}\widetilde{S}\widetilde{V}^\dagger$, then $ZQ = \widetilde{U}\widetilde{S}(UV^\dagger\widetilde{V})^\dagger$, $Z_2 = \widetilde{U}_r\widetilde{S}_r\widetilde{V}_r^\dagger$, and $Z_1 = \widetilde{U}_r\widetilde{S}_r(UV^\dagger\widetilde{V})_r^\dagger = \widetilde{U}_r\widetilde{S}_r\widetilde{V}_r^\dagger(UV^\dagger)^\dagger$. It is easy to check that $Z_1Q = Z_2P^\dagger$. $\square$

**Theorem 3.** *Assume $\rho_{\mathcal{X}}$ is unitary. Then the global optima in the function space with data augmentation and the global optima in the constrained function space are the same, i.e., $\widehat{W}^{da} = \widehat{W}^{inv}$.*

*Proof.* First, we want to prove that

$$|\mathcal{G}|\overline{G} \left( \sum_{g \in \mathcal{G}} \rho_{\mathcal{X}}(g)XX^{\mathrm{T}}\rho_{\mathcal{X}}(g)^{\mathrm{T}} \right)^{-1} = P^{-1} \left( \mathbf{I}_{d_0} - \left( P^{-1}G \right) \left( P^{-1}G \right)^{+} \right) P^{-1} \tag{47}$$

Similar to the proof of Proposition 3, we know that $\left( \sum_{g \in \mathcal{G}} \rho_{\mathcal{X}}(g)XX^{\mathrm{T}}\rho_{\mathcal{X}}(g)^{\mathrm{T}} \right)^{-1}$ commutes with $\rho_{\mathcal{X}}(g)$ for all $g \in \mathcal{G}$. Then, $\left( \sum_{g \in \mathcal{G}} \rho_{\mathcal{X}}(g)XX^{\mathrm{T}}\rho_{\mathcal{X}}(g)^{\mathrm{T}} \right)^{-1}$ commutes with $\overline{G}$ as well. According to Lemma 3, $Q^{-1} = \left( \sum_{g \in \mathcal{G}} \rho_{\mathcal{X}}(g)XX^{\mathrm{T}}\rho_{\mathcal{X}}(g)^{\mathrm{T}} \right)^{-\frac{1}{2}}$ commutes with $\overline{G}$. We also know that

$|\mathcal{G}|\overline{G}\left(\sum_{g\in\mathcal{G}}\rho_{\mathcal{X}}(g)XX^{\mathrm{T}}\rho_{\mathcal{X}}(g)^{\mathrm{T}}\right)^{-1}$ is a $\mathcal{G}$-fixed point. Therefore, we have

$$|\mathcal{G}|\overline{G}\left(\sum_{g\in\mathcal{G}}\rho_{\mathcal{X}}(g)XX^{\mathrm{T}}\rho_{\mathcal{X}}(g)^{\mathrm{T}}\right)^{-1}=|\mathcal{G}|\overline{G}\left(\sum_{g\in\mathcal{G}}\rho_{\mathcal{X}}(g)XX^{\mathrm{T}}\rho_{\mathcal{X}}(g)^{\mathrm{T}}\right)^{-1}\overline{G}$$

$$=|\mathcal{G}|\overline{G}Q^{-1}Q^{-1}\overline{G}=|\mathcal{G}|\overline{G}Q^{-1}\overline{G}Q^{-1}=(|\mathcal{G}|^{\frac{1}{2}}\overline{G}Q^{-1})^2.$$

On the other hand, $\mathbf{I}_{d_0}-(P^{-1}G)(P^{-1}G)^+$ is an idempotent projection matrix. Therefore, we have

$$P^{-1}\left(\mathbf{I}_{d_0}-(P^{-1}G)(P^{-1}G)^+\right)P^{-1}$$

$$=P^{-1}\left(\mathbf{I}_{d_0}-(P^{-1}G)(P^{-1}G)^+\right)P^{-1}=P^{-1}\left(\mathbf{I}_{d_0}-(P^{-1}G)(P^{-1}G)^+\right)^2P^{-1}$$

$$=P^{-1}\left(\mathbf{I}_{d_0}-(P^{-1}G)(P^{-1}G)^+\right)\left(P^{-1}\left(\mathbf{I}_{d_0}-(P^{-1}G)(P^{-1}G)^+\right)\right)^\dagger$$

If Equation 47 holds, then we can apply Lemma 7 directly to get the result. Therefore, we only need to prove Equation 47.

Let $\rho_{\mathcal{X}}(g)=V\Lambda_g V^{-1}$ be the eigen-decomposition of $\rho_{\mathcal{X}}(g)$, where $g$ is the generator of $\mathcal{G}$ and $\Lambda_g$ is a diagonal matrix with the eigenvalues of $\rho_{\mathcal{X}}(g)$ on the diagonal. This can be done according to Lemma 5. Furthermore, under the assumption that $\rho_{\mathcal{X}}$ is unitary, we have $V^{-1}=V^\dagger$. It is worth noting that $\Lambda_g$ is a diagonal matrix with $|\mathcal{G}|$-th roots of unity on the diagonal, and among the $|\mathcal{G}|$-th roots of unity, $d$ of them are 1. Without loss of generality, we assume that the first $d$ eigenvalues are 1. Define $\widetilde{X}=V^{-1}X$, and let $\widetilde{X}_{1:d}$ be the first $d$ rows of $\widetilde{X}$, and $\widetilde{X}_{(d+1):d_0}$ be the last $d_0-d$ rows of $\widetilde{X}$. Now, let's simplify the LHS of Equation 47:

$$\overline{G}=\frac{1}{|\mathcal{G}|}\sum_{h\in\mathcal{G}}\rho_{\mathcal{X}}(h)=\frac{1}{|\mathcal{G}|}V\left(\sum_{h\in\mathcal{G}}\Lambda_h\right)V^{-1}$$

$$=V\left(\frac{1}{|\mathcal{G}|}\sum_{i\in[\mathcal{G}]}\Lambda_g^i\right)V^{-1}=V\begin{bmatrix}\mathbf{I}_d & 0_{d,d_0-d}\\0_{d_0-d,d} & 0_{d_0-d,d_0-d}\end{bmatrix}V^{-1}, \tag{48}$$

The last equality in Equation 48 holds because the partial geometric series to order $|\mathcal{G}|$ is 0 for any root of unity other than 1, i.e., $\sum_{j=1}^{|\mathcal{G}|}(e^{\frac{2\pi ki}{|\mathcal{G}|}})^j=0$ for any $k\neq 0$. On the other hand,

$$\sum_{g\in\mathcal{G}}\frac{1}{|\mathcal{G}|}\rho_{\mathcal{X}}(g)XX^{\mathrm{T}}\rho_{\mathcal{X}}(g)^{\mathrm{T}} \tag{49}$$

$$=\sum_{g\in\mathcal{G}}\frac{1}{|\mathcal{G}|}\rho_{\mathcal{X}}(g)XX^\dagger\rho_{\mathcal{X}}(g)^\dagger$$

$$=V\left(\sum_{g\in\mathcal{G}}\frac{1}{|\mathcal{G}|}\Lambda_g\widetilde{X}\widetilde{X}^\dagger\Lambda_g^\dagger\right)V^{-1}$$

$$=V\left(\left(\sum_{g\in\mathcal{G}}\frac{1}{|\mathcal{G}|}\mathrm{diag}(\Lambda_g)\mathrm{diag}(\Lambda_g)^\dagger\right)\odot\widetilde{X}\widetilde{X}^\dagger\right)V^{-1}$$

$$=V\left(\begin{bmatrix}1_d & 0_{d,d_0-d}\\0_{d_0-d,d} & \cdots\end{bmatrix}\odot\widetilde{X}\widetilde{X}^\dagger\right)V^{-1}$$

$$=V\begin{bmatrix}\widetilde{X}_{1:d}\widetilde{X}_{1:d}^\dagger & 0_{d,d_0-d}\\0_{d_0-d,d} & \cdots\end{bmatrix}V^{-1}. \tag{50}$$

Therefore, the LHS of Equation 47 is

$$\overline{G}\left(\sum_{g\in\mathcal{G}}\frac{1}{|\mathcal{G}|}\rho_{\mathcal{X}}(g)XX^{\mathrm{T}}\rho_{\mathcal{X}}(g)^{\mathrm{T}}\right)^{-1} \tag{51}$$

$$= V \begin{bmatrix} \mathbf{I}_d & 0_{d,d_0-d} \\ 0_{d_0-d,d} & 0_{d_0-d,d_0-d} \end{bmatrix} \begin{bmatrix} \widetilde{X}_{1:d}\widetilde{X}_{1:d}^\dagger & 0_{d,d_0-d} \\ 0_{d_0-d,d} & \cdots \end{bmatrix}^{-1} V^{-1}$$

$$= V \begin{bmatrix} \left(\widetilde{X}_{1:d}\widetilde{X}_{1:d}^\dagger\right)^{-1} & 0_{d,d_0-d} \\ 0_{d_0-d,d} & 0_{d_0-d,d_0-d} \end{bmatrix} V^{-1}. \tag{52}$$

The RHS of Equation 47 is

$$P^{-1}\left(\mathbf{I}_{d_0} - (P^{-1}G)(P^{-1}G)^+\right)P^{-1}$$

$$= V\widetilde{P}^{-1}V^{-1}\left(\mathbf{I}_{d_0} - \left(V\widetilde{P}^{-1}(\Lambda_g - \mathbf{I}_{d_0})V^{-1}\right)\left(V\widetilde{P}^{-1}(\Lambda_g - \mathbf{I}_{d_0})V^{-1}\right)^+\right)V\widetilde{P}^{-1}V^{-1}$$

$$= V\widetilde{P}^{-1}\left(\mathbf{I}_{d_0} - \left(\widetilde{P}^{-1}(\Lambda_g - \mathbf{I}_{d_0})\right)\left(\widetilde{P}^{-1}(\Lambda_g - \mathbf{I}_{d_0})\right)^+\right)\widetilde{P}^{-1}V^{-1}, \tag{53}$$

where $\widetilde{P}^2 = \widetilde{X}\widetilde{X}^\dagger$.

To prove that the LHS equals the RHS, we need to show that

$$\begin{bmatrix} \left(\widetilde{X}_{1:d}\widetilde{X}_{1:d}^\dagger\right)^{-1} & 0_{d,d_0-d} \\ 0_{d_0-d,d} & 0_{d_0-d,d_0-d} \end{bmatrix} = \widetilde{P}^{-1}\left(\mathbf{I}_{d_0} - \left(\widetilde{P}^{-1}(\Lambda_g - \mathbf{I}_{d_0})\right)\left(\widetilde{P}^{-1}(\Lambda_g - \mathbf{I}_{d_0})\right)^+\right)\widetilde{P}^{-1}. \tag{54}$$

We can see that

$$\widetilde{P}^2\left(\widetilde{P}^{-2} - \begin{bmatrix} \left(\widetilde{X}_{1:d}\widetilde{X}_{1:d}^\dagger\right)^{-1} & 0_{d,d_0-d} \\ 0_{d_0-d,d} & 0_{d_0-d,d_0-d} \end{bmatrix}\right) \tag{55}$$

$$= \mathbf{I}_{d_0} - \widetilde{P}^2\begin{bmatrix} \left(\widetilde{X}_{1:d}\widetilde{X}_{1:d}^\dagger\right)^{-1} & 0_{d,d_0-d} \\ 0_{d_0-d,d} & 0_{d_0-d,d_0-d} \end{bmatrix}$$

$$= \mathbf{I}_{d_0} - \begin{bmatrix} \widetilde{X}_{1:d}\widetilde{X}_{1:d}^\dagger & \widetilde{X}_{1:d}\widetilde{X}_{(d+1):d_0}^\dagger \\ \widetilde{X}_{(d+1):d_0}\widetilde{X}_{1:d}^\dagger & \widetilde{X}_{(d+1):d_0}\widetilde{X}_{(d+1):d_0}^\dagger \end{bmatrix}\begin{bmatrix} \left(\widetilde{X}_{1:d}\widetilde{X}_{1:d}^\dagger\right)^{-1} & 0_{d,d_0-d} \\ 0_{d_0-d,d} & 0_{d_0-d,d_0-d} \end{bmatrix}$$

$$= \mathbf{I}_{d_0} - \begin{bmatrix} \mathbf{I}_d & 0_{d,d_0-d} \\ \widetilde{X}_{(d+1):d_0}\widetilde{X}_{1:d}^\dagger\left(\widetilde{X}_{1:d}\widetilde{X}_{1:d}^\dagger\right)^{-1} & 0_{d_0-d,d_0-d} \end{bmatrix}$$

$$= \begin{bmatrix} 0_{d,d} & 0_{d,d_0-d} \\ -\widetilde{X}_{(d+1):d_0}\widetilde{X}_{1:d}^\dagger\left(\widetilde{X}_{1:d}\widetilde{X}_{1:d}^\dagger\right)^{-1} & \mathbf{I}_{d_0-d} \end{bmatrix}. \tag{56}$$

On the other hand, we rewrite $\widetilde{P}^{-1}$ block-wisely, i.e., $\widetilde{P}^{-1} = \begin{bmatrix} \widetilde{P}_{11} & \widetilde{P}_{12} \\ \widetilde{P}_{12}^\dagger & \widetilde{P}_{22} \end{bmatrix}$. By Lemma 6, we have

$$(\Lambda_g - \mathbf{I}_{d_0})\left(\widetilde{P}^{-1}(\Lambda_g - \mathbf{I}_{d_0})\right)^+\widetilde{P}^{-1} \tag{57}$$

$$= \begin{bmatrix} 0_{d,d} & 0_{d,d_0-d} \\ (\widetilde{P}_{22}^2 + \widetilde{P}_{12}^\dagger\widetilde{P}_{12})^{-1}\widetilde{P}_{12}^\dagger & (\widetilde{P}_{22}^2 + \widetilde{P}_{12}^\dagger\widetilde{P}_{12})^{-1}\widetilde{P}_{22} \end{bmatrix}\begin{bmatrix} \widetilde{P}_{11} & \widetilde{P}_{12} \\ \widetilde{P}_{12}^\dagger & \widetilde{P}_{22} \end{bmatrix}$$

$$= \begin{bmatrix} 0_{d,d} & 0_{d,d_0-d} \\ (\widetilde{P}_{22}^2 + \widetilde{P}_{12}^\dagger\widetilde{P}_{12})^{-1}(\widetilde{P}_{12}^\dagger\widetilde{P}_{11} + \widetilde{P}_{22}\widetilde{P}_{12}^\dagger) & \mathbf{I}_{d_0-d} \end{bmatrix} \tag{58}$$

By definition, we know that $\widetilde{P}^{-2}\widetilde{X}\widetilde{X}^\dagger = \mathbf{I}_{d_0}$. Therefore,

$$\begin{bmatrix} \widetilde{P}_{11} & \widetilde{P}_{12} \\ \widetilde{P}_{12}^\dagger & \widetilde{P}_{22} \end{bmatrix}^2\begin{bmatrix} \widetilde{X}_{1:d}\widetilde{X}_{1:d}^\dagger & \widetilde{X}_{1:d}\widetilde{X}_{(d+1):d_0}^\dagger \\ \widetilde{X}_{(d+1):d_0}\widetilde{X}_{1:d}^\dagger & \widetilde{X}_{(d+1):d_0}\widetilde{X}_{(d+1):d_0}^\dagger \end{bmatrix} = \mathbf{I}_{d_0}, \tag{59}$$

$$\begin{bmatrix} \widetilde{P}_{11}^2 + \widetilde{P}_{12}\widetilde{P}_{12}^\dagger & \widetilde{P}_{11}\widetilde{P}_{12} + \widetilde{P}_{12}\widetilde{P}_{22} \\ \widetilde{P}_{12}^\dagger\widetilde{P}_{11} + \widetilde{P}_{22}\widetilde{P}_{12}^\dagger & \widetilde{P}_{22}^2 + \widetilde{P}_{12}^\dagger\widetilde{P}_{12} \end{bmatrix}\begin{bmatrix} \widetilde{X}_{1:d}\widetilde{X}_{1:d}^\dagger & \widetilde{X}_{1:d}\widetilde{X}_{(d+1):d_0}^\dagger \\ \widetilde{X}_{(d+1):d_0}\widetilde{X}_{1:d}^\dagger & \widetilde{X}_{(d+1):d_0}\widetilde{X}_{(d+1):d_0}^\dagger \end{bmatrix} = \mathbf{I}_{d_0}. \tag{60}$$

By equating the LHS and RHS of the above equation, we can get that

$$(\widetilde{P}_{12}^\dagger\widetilde{P}_{11} + \widetilde{P}_{22}\widetilde{P}_{12}^\dagger)\widetilde{X}_{1:d}\widetilde{X}_{1:d}^\dagger + (\widetilde{P}_{22}^2 + \widetilde{P}_{12}^\dagger\widetilde{P}_{12})\widetilde{X}_{(d+1):d_0}\widetilde{X}_{1:d}^\dagger = 0_{d_0-d,d},$$

$$-\widetilde{X}_{(d+1):d_0}\widetilde{X}_{1:d}^\dagger\left(\widetilde{X}_{1:d}\widetilde{X}_{1:d}^\dagger\right)^{-1} = (\widetilde{P}_{22}^2 + \widetilde{P}_{12}^\dagger\widetilde{P}_{12})^{-1}(\widetilde{P}_{12}^\dagger\widetilde{P}_{11} + \widetilde{P}_{22}\widetilde{P}_{12}^\dagger) \tag{61}$$

We have shown that the LHS equals the RHS in Equation 47. The theorem is proved. $\square$

### A.8 PROOF OF PROPOSITION 4

**Proposition 7.** *Suppose the target matrix* $Z \in \mathbb{R}^{d_L \times d_0}$ *has rank* $m > d > r$. *The critical points of* $\ell_Z$ *restricted to the function space* $\mathcal{M}_r$ *are all matrices of the form* $U\Sigma_\mathcal{I}V^{\mathrm{T}}$ *where* $\mathcal{I} \in [d]_r$. *If* $0 < \sigma_{r+1} < \sigma_r$, *then the local minimum is the critical point with* $\mathcal{I} = [r]$. *It is the global minimum.*

The proof is adapted from the proof of (Trager et al., 2020, Theorem 28).

*Proof.* A matrix $P \in \mathcal{M}_r$ is a critical point if and only if $Z - P \in N_P\mathcal{M}_r = \mathrm{Col}(P)^\perp \otimes \mathrm{Row}(P)^\perp$, where $N_P\mathcal{M}_r$ denotes the normal space of $\mathcal{M}_r$ at point $P$. If $P = \sum_{i=1}^r \sigma_i' (u_i' \otimes v_i')$ and $Z - P = \sum_{j=1}^e \sigma_j'' (u_j'' \otimes v_j'')$ are SVD with $\sigma_i' \neq 0$ and $\sigma_j'' \neq 0$, the column spaces of $P$ and $Z - P$ are spanned by the $u_i'$ and $u_j''$, respectively. Similarly, the row spaces of $P$ and $Z - P$ are spanned by the $v_i'$ and $v_j''$, respectively. So $P$ is a critical point if and only if the vectors $u_i', u_j''$ and $v_i', v_j''$ are orthonormal, i.e., if

$$Z = P + (Z - P) = \sum_{i=1}^r \sigma_i' (u_i' \otimes v_i') + \sum_{j=1}^e \sigma_j'' (u_j'' \otimes v_j'')$$

is a SVD of $Z$. This proves that the critical points are of the form $U\Sigma_\mathcal{I}V^{\mathrm{T}}$ where $Z = U\Sigma V^{\mathrm{T}}$ is a SVD and $\mathcal{I} \in [d]_r$. Since $\ell_Z\left(U\Sigma_\mathcal{I}V^{\mathrm{T}}\right) = \left\|U\Sigma_{[d]\setminus\mathcal{I}}V^{\mathrm{T}}\right\|^2 = \left\|\Sigma_{[d]\setminus\mathcal{I}}\right\|^2 = \sum_{i\notin\mathcal{I}} \sigma_i^2$, we see that the global minima are exactly the critical points selecting $r$ of the largest singular values of $Z$, i.e., with $\mathcal{I} = [r]$. It is left to show that there are no other local minima. For this, we consider a critical point $P = U\Sigma_\mathcal{I}V^{\mathrm{T}}$ such that at least one selected singular value $\sigma_i$ for $i \in \mathcal{I}$ is strictly smaller than $\sigma_r$. This is possible since $0 < \sigma_{r+1} < \sigma_r$. To see that $P$ cannot be a local minimum, one can follow the proofs in (Trager et al., 2020, Theorem 28). $\square$

**Proposition 4.** *Assume all non-zero singular values of* $\overline{Z}^{inv}, \overline{Z}^{da}, \overline{Z(\lambda)}^{reg}$ *are pairwise distinct.*

1. *(Constrained Space) The number of critical points in the optimization problem (4) is* $\binom{d}{r}$. *They are all in the form of* $\overline{U}^{inv}\overline{\Sigma}_\mathcal{I}^{inv}\overline{V}^{inv\,\mathrm{T}}P^{-1}$, *where* $\mathcal{I} \in [d]_r$. *The unique global minimum is* $\overline{U}^{inv}\overline{\Sigma}_{[r]}^{inv}\overline{V}^{inv\,\mathrm{T}}P^{-1}$, *which is also the unique local minimum.*

2. *(Data Augmentation) The number of critical points in the optimization problem (9) is* $\binom{d}{r}$. *They are all in the form of* $\overline{U}^{da}\overline{\Sigma}_\mathcal{I}^{da}\overline{V}^{da\,\mathrm{T}}Q^{-1}$, *where* $\mathcal{I} \in [d]_r$. *These critical points are the same as the critical points in the constrained function space. The unique global minimum is* $\overline{U}^{da}\overline{\Sigma}_{[r]}^{da}\overline{V}^{da\,\mathrm{T}}Q^{-1}$, *which is also the unique local minimum.*

3. *(Regularization) The number of critical points in the optimization problem (7) is* $\binom{m}{r}$. *They are all in the form of* $\overline{U}^{reg}\overline{\Sigma}_\mathcal{I}^{reg}\overline{V}^{reg\,\mathrm{T}}B(\lambda)^{-1}P^{-1}$, *where* $\mathcal{I} \in [m]_r$. *The unique global minimum is* $\overline{U}^{reg}\overline{\Sigma}_{[r]}^{reg}\overline{V}^{reg\,\mathrm{T}}B(\lambda)^{-1}P^{-1}$, *which is also the unique local minimum.*

*Proof.* This follows directly from Proposition 7 and the fact that $\overline{Z}^{da}$ and $\overline{Z}^{inv}$ are both rank $d$ matrices while $\overline{Z}^{reg}$ has rank $m$. $\square$

### A.9 EMPIRICAL SPECTRUM OF TARGET MATRICES IN MNIST DATASET

As discussed in Remark 2 and Proposition 4, we have assumptions about the rank and spectrum of the target matrices we are trying to approximate. As shown in Figure 4, we empirically computed the singular values of $\overline{Z}^{da}$, $\overline{Z}^{inv}$, $\overline{Z(\lambda)}^{reg}$ for MNIST dataset. We can see that all three target matrices have full rank. The singular values are pairwise different as well. Thus, the previous assumptions in Remark 2 and Proposition 4 are satisfied.

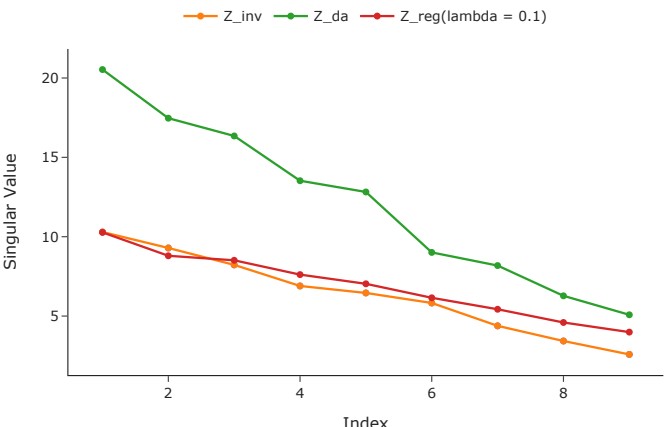

Figure 4: The spectrum of target matrices in the MNIST dataset.

### A.10 COMPARISON BETWEEN DATA AUGMENTATION AND REGULARIZATION UNDER CROSS ENTROPY LOSS

In Figure 5, we are still plotting $\|W^{\perp}\|_F$ for data augmentation and regularization trained on the same dataset, but with cross entropy loss. It is observed that, for larger $\lambda$, the dynamics of $\|W^{\perp}\|_F$ resemble those when trained with MSE (see Figure 3). On the other hand, for small $\lambda$, $\|W^{\perp}\|_F$ may increase at first, and then decrease. For data augmentation, if we allow more epochs, we can still observe that $\|W^{\perp}\|_F$ decreases after increasing. Our theoretical results only support the scenario for mean squared loss. Thus, when trained with cross entropy, we cannot say whether all the critical points are invariant or not. Future work can be done to investigate the critical points when trained with cross entropy loss.

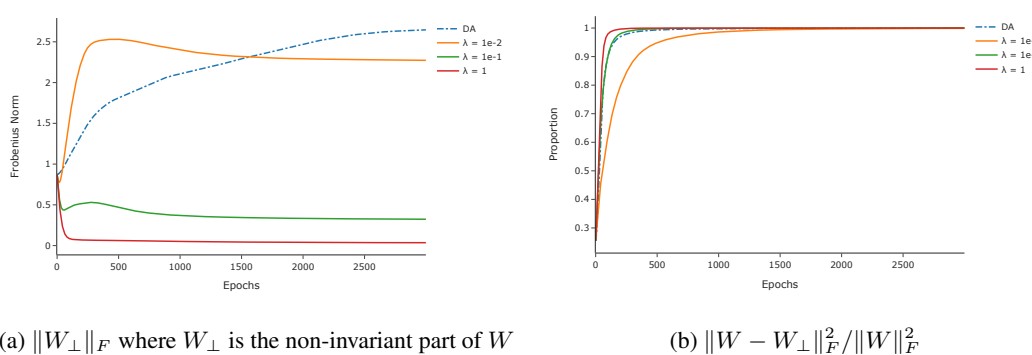

(a) $\|W_{\perp}\|_F$ where $W_{\perp}$ is the non-invariant part of $W$

(b) $\|W - W_{\perp}\|_F^2 / \|W\|_F^2$

Figure 5: Frobenius norm of the non-invariant part of the end-to-end matrix $W$, trained with Data Augmentation (DA) and Regularization ($\lambda$) using Cross Entropy Loss (CE).

## A.11 EXPERIMENTS FOR TWO-LAYER NONLINEAR NETWORK

In Figure 6, we show the training curve for a two-layer neural network with different nonlinear activation functions trained with data augmentation and hard-wiring. The setup is the same as previous experiments in section 4. In this experiment, we used 5000 samples from the MNIST dataset for training with mean squared loss (MSE). Meanwhile, we also test the case when there is not a bottleneck middle layer. When the middle layer has a bottleneck, we set the number of hidden units as 7; otherwise, the number of hidden units is 15. We can see that both data augmentation and constrained model have similar loss in the late phase of training for all four activation functions, especially when there is a bottleneck.

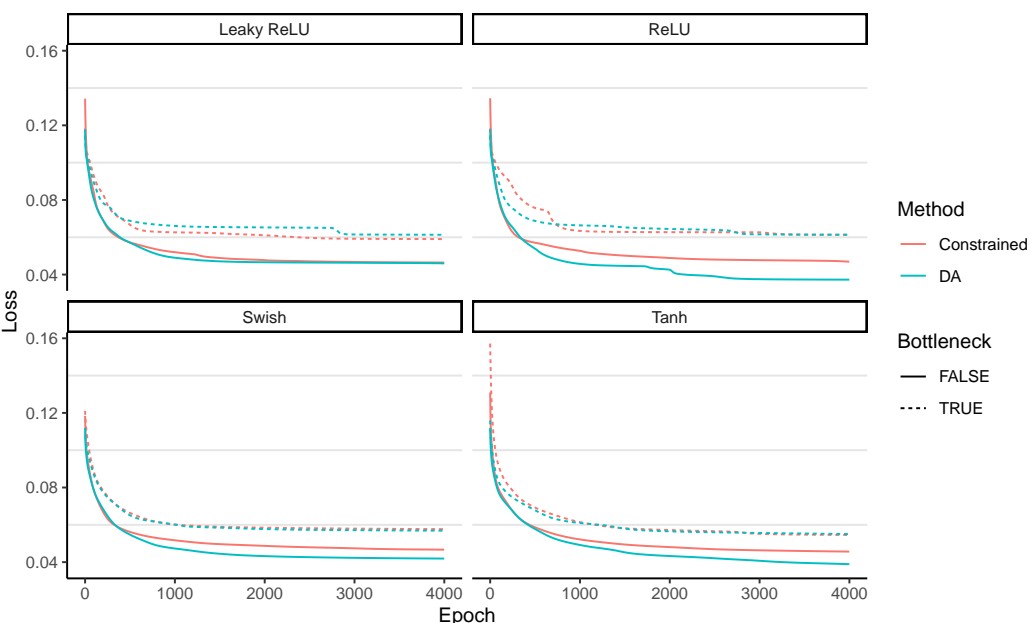

Figure 6: Training curves for a two-layer NN with different nonlinear activation functions via data augmentation and hard-wiring on MNIST.

Besdies, Moskalev et al. (2023) suggests that invariance learned from data augmentation deteriorates under distribution shift in a classification setting. The architecture they choose is a 5-layer ReLU network. We would like to investigate this in a regression setting. The model we use is a 2-layer neural network with different activation functions trained with data augmentation and MSE. For any function $f$ and an input point $x \in \mathcal{X}$, to measure the amount of invariance of $f$, we evaluate the scaled variance of outputs across the group orbit,

$$\epsilon_{inv}(f, x) := \mathbb{E}_{g \sim \lambda} \left( 1 - \frac{f(gx)}{\overline{f}(x)} \right)^2,$$

where $\overline{f}(x) := \mathbb{E}_{g \sim \lambda}[f(gx)]$, and $\lambda$ is the Haar measure on group $\mathcal{G}$. In Figure 7, we train a 2-layer neural network with different activation functions on MNIST with data augmentation using training sample sizes $N = 1000$ and $N = 5000$. After training, we calculate $\epsilon_{inv}(f, x)$ for two different datasets: MNIST and Gaussian. For MNIST, we use 5000 samples from the original test set in MNIST. For Gaussian, we sample 5000 points from an isotropic Gaussian distribution in dimension 196. We are showing the median of $\{\epsilon_{inv}(f, x)\}_{x \in \mathcal{D}}$ in Figure 7.

The observations can be summarized as follows:

1. **Effect of the size of the model:** Compared to the case without a bottleneck middle layer, $\epsilon_{inv}(f, x)$ is significantly smaller when there is a bottleneck. This suggests that it is more difficult to learn invariance from the data when the model has more parameters.

2. **Effect of the amount of training data:** We notice that $\epsilon_{inv}(f,x)$ is smaller when there are more training data. For underdetermined linear models, i.e., when the number of data points exceeds the input dimension, Proposition 4 shows that all critical points are invariant. However, when the model is nonlinear, we need more data in order to learn the invariance via data augmentation.

3. **Robustness under distribution shift:** Though the model is trained on MNIST, $\epsilon_{inv}(f,x)$ does not increase significantly even when the model is tested on a completely different dataset. This suggests that the invariance learned from the data is fairly robust.

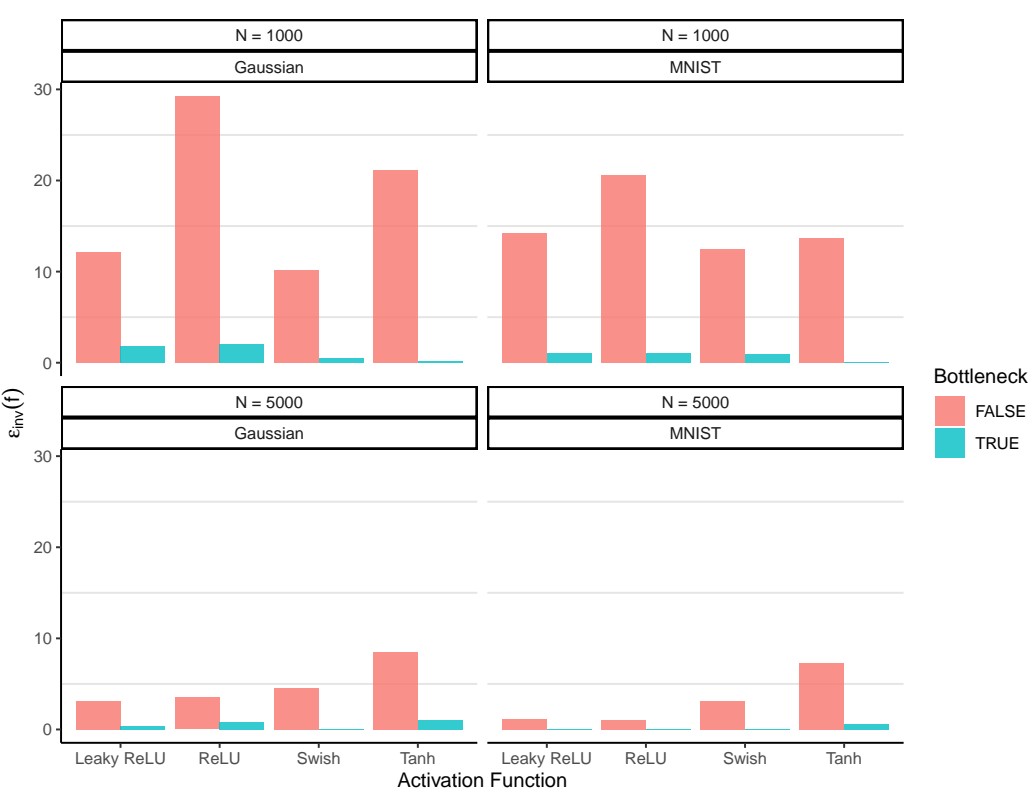

Figure 7: Median of the measure $\epsilon_{inv}(f)$ of discrepancy from invariance for 2-layer neural networks with different activation functions, trained on MNIST with data augmentation.

