# OpenReview forum: "Geometry of the Loss Landscape in Invariant Deep Linear Neural Networks"
_ICLR.cc/2025/Conference — Submitted to ICLR 2025_

### Official Review · Reviewer_NSSs · 2024-10-19

**Soundness:** 3
**Presentation:** 2
**Contribution:** 2
**Rating:** 6
**Confidence:** 4

**Summary:**

This paper investigates how different methods of implementing invariance - by having it hard wired, imbued in the data, or encouraged using regularisation - affect optimisation. They study the simple case of deep linear networks, rank-constraining them to make the optimisation non-convex. In this case the three different settings share the same optimum in the regulariser’s limit and have the same critical points, with the regularisation having more than the hard-wiring/data-augmentation cases.

**Strengths:**

**Novelty:**

This work studies an important, timely question - the relation between hard-wiring and learning symmetries. Although not explicitly stated, this would have implications for network design and generally when should one hardwire a symmetry vs just imbue it in the data and whether if there is a hidden symmetry will it be learnt.

**Scientific quality:**

The setting nicely relates between the three different cases, albeit naturally in the limited linear case given here.

**Clarity:**

The paper is well written and neatly organised. It has a good, logical flow, giving examples and generally trying to expand on theorems where possible. Section 2 is especially well, concisely presented given the large background.

**Weaknesses:**

**Novelty:**

As there are many works trying to answer this question it’s currently unclear what this one adds that others don’t. It’s known that networks can learn to respect symmetries, with empirical results being given eg. in [5-6]. [2] looked at using data augmentations vs feature averaging, which is not the same but similar to the data augmentation vs hardwiring cases here. These are select examples but still, the contrast to existing works isn’t as clear as it would ideally be.

**Scientific quality:**

It’s unclear how the rank-constrained setting is related to real cases. In practice networks are universal - linear networks are often assumed for tractability but is limiting the expressivity realistic, and if so then how? Is it assumed solely to make the loss landscape nonconvex and if so, why not get that through a myriad of other ways? Recommend clarifying this.

There’s a special focus on cyclic and finite groups, without a clear explanation/motivation why. Do these encompass many of the common groups in geometric deep learning? What do they fail to describe?

Lines 397-399 - recommend showing how some weights tending to the same value results from the theorems.

**Clarity:**

This paper suffers from the page limit as many similar theoretical deep learning papers do. This stops the authors from sufficiently expanding on the different results and giving more intuition for their theorems. Still, currently there’s an insufficient emphasis on the “why” relative to the “what”. For example, the abstract details the problem, the setting, and the results, but not what they mean, hence not explicitly answering the problem. This problem is evident on many levels in the main body as well. This is evident also in the contributions section (1.1). Other than that high level, theorems are given with little intuition as to why they hold or what they mean. The paper has a decent information density as it is so it’s naturally difficult to accommodate everything, but it can not only leave the reader confused but more importantly make it harder to understand how the results tie together and get at the underlying problem. Another example is the related work section - it’s unclear what important context these works give relating to this study, and if they don’t then why they are mentioned at all.

Throughout the paper numbers/axis titles on plots should be bigger.

Lines 47-49 - The stated problem is different than the impression one gets from the abstract - the latter implies “can symmetries be learnt and how does their optimisation look” whereas the former says something else. Recommend rephrasing either or both.

Line 53 - what does benevolent mean here? Recommend replacing.

Line 71-72 - unclear what’s meant here by linear invariant neural networks given the different settings.

Lines 197 - shouldn’t det-variety be defined? It’s not a well known term, and if it’s not important enough to be defined it should be delegated to the appendix or not used.

Line 211 - what’s r in this context?

**Questions:**

**Scientific quality:**

Lines 74-76 - how is this intuitive/obvious? It’s important to give that intuition. To play devil’s advocate, if it’s obvious then it’s even moreso important to clarify why this is studied and what new insight was achieved if any.

Lines 234-236 - isn’t it possible to formulate this for any group with a countable number of generators?

Lines 243-245 - do all roots work equally well? Assuming this corresponds to several classes of solutions no?

Lines 267-269, remark 2 - this is a good example but it’s unclear how typical this is, although it makes some intuitive sense that it would be. Making that clearer could be nice, are there more grounded reasons to believe some analogy of this generally holds? Seems related to the manifold hypothesis - you’re assuming there’s some latent structures and small deviations from it. Also although the rank can technically be large it might have many small singular values, no?

Line 395-396 - why change Adam’s betas?

Figure 1 - it’s quite interesting how results are similar for CE even though the theorems don’t hold for it, is this discussed anywhere?

For the hard-wiring experiments why not use a different B with a different size? Eg. to make all cases have a similar number of parameters, although their expressivity is clearly the same.

Lines 486-496 - this is a nice decomposition but I believe I saw it in other works, are you aware of it appearing elsewhere in the literature?

Line 496-497 - why is the double descent interesting? Do you mean the orange line in figure 3.a?

**Clarity:**

Line 41 - “have shown” how? Feels like a citation’s needed.

Lines 47-49 - where in the paper are the solutions studied/referred to? Eg. when are the regularised solutions invariant? This is implicitly shown but not discussed.

Lines 123-124 - deteriorates how so? This is interesting and seems relevant here, consider slightly expanding.

Lines 145-146 - why is the tangent space relevant here? I didn’t understand where it was heavily used throughout the paper.

Lines 162-170 - missing caption?

Lines 169-170 - is finite and cyclic defined anywhere?

Lines 285-293 - the connection to manifold regularisation is interesting but it’s unclear what it adds - what’s lost if it’s removed? What does it say in this context?

Lines 295,296 - isn’t \bar{Z(\lambda)}^{reg} defined twice? Is it just different forms of the same expression? Generally this theorem’s intuitive meaning/interpretation is unclear.

Lines 303-305, thm 2 - why wouldn’t it be continuous, due to the rank constraint? Generally solutions to L2 regularisations are continuous so recommend making this clearer.

Lines 333-336 - this is quite interesting, it would be nice to discuss this - why it happens, potential implications, etc.

Many of the previous kinds of comments about intuition/interpretation and what vs why hold for section 3.4 as well.

Lines 373-377 - does spurious here mean suboptimal? Recommend clarifying.

**Minor points, suggestions, etc.:**

In the first paragraph what about classical examples eg. graphs/images?

There are some papers that weren’t mentioned which could be relevant throughout the paper, eg. [1-3, 7]. [2] specifically has some potential overlap and I recommend clarifying what’s different than their work. [7] might have overlap with section 4.3, specifically the weight decomposition.

Recommend merging section 1.1 with 1.

[4] and generally Saxe/Ganguli’s works are relevant both in spirit and regarding results, at least as some of the first to study deep linear networks in modern deep learning.

Lines 270-274 basically say that you’re taking projections and as everything is linear it’s fine, which is nice but can be spelled out more explicitly.

Section 3.3 - can anything meaningful be said about the case where the symmetry is only “on average” embedded in the data, so only partial group orbits are included?

Some small experiments with regular nonlinear networks showing whether these results hold and if so then to what extent would be instructive.

[1] Gerken, Jan E., and Pan Kessel. "Emergent Equivariance in Deep Ensembles." arXiv preprint arXiv:2403.03103 (2024).

[2] Lyle, Clare, et al. "On the benefits of invariance in neural networks." arXiv preprint arXiv:2005.00178 (2020).

[3] Fuchs, Fabian Bernd. Learning invariant representations in neural networks. Diss. University of Oxford, 2021.

[4] Saxe, Andrew M., James L. McClelland, and Surya Ganguli. "Exact solutions to the nonlinear dynamics of learning in deep linear neural networks." arXiv preprint arXiv:1312.6120 (2013).

[5] Olah, C., Cammarata, N., Voss, C., Schubert, L., and Goh, G. Naturally occurring equivariance in neural networks. Distill, 2020. doi: 10.23915/distill.00024.004. https://distill.pub/2020/circuits/equivariance.

[6] Gruver, Nate et al. (2023). “The Lie Derivative for Measuring Learned Equivariance”.
In: The Eleventh International Conference on Learning Representations. url: https:
//openreview.net/forum?id=JL7Va5Vy15J.

[7] Gideoni, Yonatan. "Implicitly Learned Invariance and Equivariance in Linear Regression."

**Decision:**
As it is I recommend rejecting this paper mostly on the grounds of clarity, but that’s assuming that it presents a deeper novelty than what it currently seems to. It’s unclear if it has meaningful implications for real networks but it might still be insightful to consider this toy case, although this remains to be seen.

---

**Update post-discussions:**

Following a thorough discussion with the authors and them addressing the main comments regarding clarity I am raising my score to 6 and recommend accepting this paper. This is a high quality work where the authors investigate three different settings of a relevant problem that is generally considered open in the geometric deep learning community. The paper's main downside is that of many theoretical deep learning works where theoretical insights are insufficiently tied back to more realistic settings, although the revised manuscript minimises this gap as much as possible without additional experiments. This is a shame as even simple MNIST-esque experiments would go a long way. Still, I believe this paper will be of interest to the community, especially if future work builds upon it.

---

> ### Author Response · Authors · 2024-11-20
>
> Thank you for your very detailed review. We have updated the manuscript in response to your review. Answers to some of the main questions you asked are provided below:
>
> **1. Line 41 - “have shown” how? Feels like a citation’s needed.**
>
> Citations have been added.
>
> **2. Lines 47-49 - where in the paper are the solutions studied/referred to? Eg. when are the regularised solutions invariant? This is implicitly shown but not discussed.**
>
> The solutions are studied in Section 3 (Section 3.1 for hard-wiring, Section 3.2 for regularization, Section 3.3 for data augmentation). The regularized solutions are not invariant when the penalty coefficient $\lambda$ is finite.
>
> **3. Lines 74-76 - how is this intuitive/obvious? It’s important to give that intuition. To play devil’s advocate, if it’s obvious then it’s even moreso important to clarify why this is studied and what new insight was achieved if any.**
>
> We stated that this might seem intuitive to some readers, but that we certainly do not consider it to be obvious. We have rephrased to avoid confusion. To explain what that intuition might be: augmenting the data across the group orbits does not bring new information other than enforcing the group symmetry. Therefore, one might expect that data augmentation should have a similar effect as constraining (hard-wiring) the model.
>
> **4. Lines 145-146 - why is the tangent space relevant here? I didn’t understand where it was heavily used throughout the paper.**
>
> It is used in proving Proposition 7. Since it is not heavily used, we have removed this line, and added it in Proposition 7.
>
> **5. Lines 197 - shouldn’t det-variety be defined? It’s not a well known term, and if it’s not important enough to be defined it should be delegated to the appendix or not used.**
>
> We added a reference in the main text. A brief description is included in accessible language. We mentioned this to emphasize how this set is different from a linear space. To explain the terminology: a matrix has rank at most $r$ if all of its sub-matrices of size $(r+1)\times(r+1)$ have determinant zero. Thus, the set of matrices of rank at most $r$ is the set of matrices whose entries satisfy a list of polynomial equations (this makes it a variety) and these equations can be written as determinants (making it a determinantal variety).
>
> **6. Line 211 - what’s r in this context?**
>
> Consider a fully-connected linear network where the narrowest layer has width $r$. The function space is the set of end-to-end matrices, which are $n \times m$ matrices of rank at most $r$. The manuscript has been updated to clarify this.
>
> **7. Lines 234-236, Finitely generated groups:**
>
> We presented our results for the case of cyclic groups or finitely generated groups. However, extending our results to continuous groups (such as $SO(n)$ and $SE(n)$) is straightforward, since Lie theory provides a way of analyzing continuous groups in terms of their infinitesimal generators. To explain this point we have updated the manuscript to include Remark 1 and an Appendix A.1 with details about the extension of our results to Lie groups.
>
> **8. Lines 243-245 - do all roots work equally well? Assuming this corresponds to several classes of solutions no?**
>
> It is known that a positive definite matrix has a unique square root.
>
> **9. Line 303-305, Continuity of regularization path:**
>
> Yes, due to the rank constraints, it is not clear whether the regularization path is continuous. We are able to show that it is indeed a continuous path.
>
> **10. Lines 373-377 does spurious here mean suboptimal? Recommend clarifying.**
>
> Pure critical points are critical points that arise from the geometry of the function space, while spurious ones result solely from the parametrization map and do not correspond to critical points in the function space.
>
> **11. Line 395-396 why change Adam’s betas?**
>
> We haven't changed betas of Adam. These are the default values in PyTorch.
>
> **12. Lines 486-496 - this is a nice decomposition but I believe I saw it in other works, are you aware of it appearing elsewhere in the literature?**
>
> It is a relatively intuitive decomposition that other people may also have come up with. We are not aware of other work using exactly the same setup as here.
>
> **13. Line 496-497 - why is the double descent interesting? Do you mean the orange line in figure 3.a?**
>
> All the lines in 3(a) have different degrees of "double descent", especially the blue (data augmentation) and the orange ones. Our conjecture is that the loss may also be decomposed into two parts, one controlling the error of invariance, and the other one controlling the error from the target. Therefore, the gradient of the weights during training can be decomposed into two directions as well, and their differences may result into this phenomenon. This can help us better understand the training dynamics of those models, which eventually could shed light on methods to accelerate training.

---

> > ### Author Response · Authors · 2024-11-20
> >
> > **14. Regarding the rank constraint:**
> >
> > The rank constraints are actually common in practice. In generative models like variational encoder (VAE), the hidden layer is usually narrower than both input and output layers, since people believe there is a low-dimensional latent representation for the data. Recently, in large language models, low-rank adaptation (LoRA) is also widely used to reduce the number of trainable parameters for downstream tasks. As long as a narrow linear layer is used in the architecture, rank constraints will arise. Thus, rank constraints are not rare in real cases. We are interested in rank constraints because the corresponding function space is not a vector space. Networks with nonlinear activation functions are also nonlinear subsets of vector spaces. The non-convexity of the function space makes nontrivial to draw conclusions about the impact of the constraints imposed by invariances. Studying nonlinear constraints in function space is an important research program, and rank constraints are in our opinion one of the most natural places to start.

---

> ### Comment · Reviewer_NSSs · 2024-11-20
>
> Thanks for the response and various clarifications, I hope they'll help future readers. Regarding some of the points, starting from minor:
>
> **3** - it might be useful giving this intuition explicitly, it's helpful.
>
> **4** - word-searching the document I can't find "tangent". I don't think this is an issue but FYI.
>
> **7** - excellent. Note that proposition 5, line 831, has "group Lie group"
>
> **8** - I believe you mean a unique _PSD_ root, no? https://mathoverflow.net/questions/266286/is-the-square-root-of-a-matrix-unique would appreciate clarifying this in the main text.
>
> **9** - I see, recommend clarifying this.
>
> **10** - Fascinating, thanks for clarifying.
>
> **11** - then why mention them? Doesn't matter much either way as you've now clarified it.
>
> **12** - likely not exactly the same but I believe [1] for example has that decomposition. It isn't a main result of 4.3 and hence I don't think detracts from it, but perhaps worth mentioning.
>
> **13** - this is an interesting conjecture, it's good you added it to the main text.
>
> And regarding more major, conceptual ones:
>
> **2** - believe my issue here is mostly the discussion part, apologies for not making it clearer. Specifically - the different kinds of solutions for each setting are found, but their implications are not deeply discussed. The solutions themselves are interesting but it is not said _why_ they are interesting, or what insight we get from them. This is related to what I wrote under **Decision**, where currently the paper's novelty is presented via its results but insufficiently interpreted and spoon-fed to the reader. Even if the results are interesting only to someone working on theoretical geometric deep learning, ideally any deep learning researcher can read the paper and understand why people in that community find it interesting.
>
> **Clarity** - the first paragraph of clarity issues mentioned in the weaknesses still for the most part hold. Small changes here and there have helped alleviate them but insufficiently. For example, the conclusion section states the results but doesn't say what's interesting about them, how they may carry over to real networks, etc. The related work section still goes over existing works without the reader always understanding why they are mentioned, or explicitly how the current work answers gaps/limitations they have - see [2] for a good example of a related work section which gives good context.
>
> **Rank constraint** - An earlier version of this comment was posted before/synchronously to your reply regarding the rank constraint. I think that explanation is extremely elucidating and implore you to add it to the paper.
>
> If I didn't comment on something it means it's a good change or I didn't notice it.
>
> [1] Gideoni, Yonatan. "Implicitly Learned Invariance and Equivariance in Linear Regression."
>
> [2] Finn, Chelsea, Pieter Abbeel, and Sergey Levine. "Model-agnostic meta-learning for fast adaptation of deep networks." International conference on machine learning. PMLR, 2017.

---

> > ### Author Response · Authors · 2024-11-22
> >
> > **1. Regarding 4 - word-searching the document I can’t find ”tangent”. I don’t think this is an issue but FYI.**
> >
> > In Proposition 7, we used ”normal space”. We don’t use ”tangent space” explicitly.
> >
> > **2. Regarding 7 - excellent. Note that proposition 5, line 831, has ”group Lie group”**
> >
> > Thank you for the catch. We have updated the manuscript to correct this typo.
> >
> > **3. Regarding 8 - I believe you mean a unique PSD root, no?**
> >
> > Yes, we mean the PSD root. We have updated the manuscript accordingly.
> >
> > **4. Regarding 9 - I see, recommend clarifying this. (regularization path)**
> >
> > We have added a comment before Theorem 2 for clarification in the updated manuscript.
> >
> > **5. Regarding 11 - then why mention them? Doesn’t matter much either way as you’ve now clarified it.**
> >
> > We added the value of the Adam parameter in the interest of ensuring reproducibility. Thank you for your comment, we think the description is more understandable with the added clarification.
> >
> > **6. Regarding 12 - likely not exactly the same but I believe [1] for example has that decomposition. It isn’t a main result of 4.3 and hence I don’t think detracts from it, but perhaps worth mentioning.**
> >
> > We have updated the manuscript to include this reference in Section 4.3.
> >
> > **7. Regarding 2**
> >
> > We have updated the conclusion section to better illustrate the implications and insights. Based on our theoretical results, we suggest that in the context of learning with invariance the data-augmented model may have similar performance as
> > the constrained model, but will incur higher data and computation costs. The regularized model does not require more data and should have a performance close to the constrained model, but it may induce more critical points than the
> > other two methods. The constrained model should have the best performance, though one might need to design the invariant architecture carefully before feeding the data to the model. Establishing this type of results in other settings would
> > be a valuable future endeavor.
> >
> > **8. Clarity**
> >
> > Your comment is well taken. As indicated above, we have updated the conclusion section to better highlight the takeaway messages and future directions. We have also updated the introduction to better explain how the model that we consider might serve as a first step to address other nonlinear models and how rank constraints are common in practice. Lastly, we have reworked the related works section to streamline the presentation of the research context and relevant references as well as more clearly explain the relations and differences to our work.
> >
> > **9. Regarding Rank Constraints**
> >
> > Thank you! We have added the explanation to the introduction section.

---

> > > ### Comment · Reviewer_NSSs · 2024-11-23
> > >
> > > Thanks for addressing all remaining comments. I will update my main review accordingly. Minor notes:
> > >
> > > - Saxe's works, eg. [1], aren't mentioned anywhere. As they are an early pioneer of studying deep linear networks they should appear somewhere in the related works or introduction sections.
> > >
> > > - At the end of 1.2 "loss landscapes" you may want to reiterate how you focus on linear networks that cannot be parameterised as linear models, unlike previous works.
> > >
> > > - Although discussing equivariance, [2] take an empirical approach which may be relevant for your related work, as their insights somewhat complement yours.
> > >
> > > [1] Saxe, Andrew M., James L. McClelland, and Surya Ganguli. "Exact solutions to the nonlinear dynamics of learning in deep linear neural networks." arXiv preprint arXiv:1312.6120 (2013).
> > >
> > > [2] Gruver, Nate, et al. "The lie derivative for measuring learned equivariance." arXiv preprint arXiv:2210.02984 (2022).

---

> > > > ### Author Response · Authors · 2024-11-24
> > > >
> > > > Thank you for providing the additional comments.
> > > >
> > > > We have updated the manuscript to add Saxe's work into the introduction. We also reiterated our focus at the end of 1.2. Regarding Gruver's work, it is indeed interesting to measure the amount of equivariance based on Lie derivative. However, since our manuscript is already compact, we may not be able to discuss their work in the paper. But we will indeed take their insights and may build future work upon it.

---

### Official Review · Reviewer_GZqx · 2024-10-27

**Soundness:** 2
**Presentation:** 2
**Contribution:** 2
**Rating:** 5
**Confidence:** 3

**Summary:**

This paper explores how different methods for enforcing invariance in neural networks—such as hard-wiring, regularization, and data augmentation—affect the loss landscape and optimization process. Focusing on deep linear networks, the authors compare these approaches in terms of their critical points and global optima. They show that for rank-constrained linear maps, both hard-wiring and data augmentation yield the same critical points, most of which are saddle points except for the global optimum. Regularization, while producing more critical points, eventually converges to the same global optima. The study provides theoretical insights into how these methods influence learning processes in machine learning models and helps explain their performance in reducing sample complexity. The authors also present experimental results to demonstrate convergence behavior in practical settings.

**Strengths:**

* The paper provides a deep theoretical comparison of different approaches to enforce invariance (data augmentation, regularization, and hard-wiring). By proving the equivalence of global optima across these methods and analyzing critical points in function space, the authors offer valuable insights into the optimization landscapes of invariant models.

* The paper specifically addresses the impact of invariance in deep linear networks, which is a significant area in machine learning. By narrowing down the study to a structured problem, it successfully derives concrete results that are applicable to broader, more complex architectures.

* The combination of theoretical results with empirical validation is a strong aspect of this paper. The authors provide experimental evidence supporting their theoretical conclusions, such as the similarity in performance and convergence rates of data augmentation and hard-wired models. This connection strengthens the practical relevance of the theoretical findings.

**Weaknesses:**

* The paper focuses exclusively on deep linear networks, which are a simplified model of neural networks. While this approach allows for clear theoretical insights, the results may not fully generalize to more complex architectures, such as non-linear or deep convolutional networks that are commonly used in real-world applications.

* The study centers on particular group-invariant structures, which might not cover a wide range of practical invariance cases. Invariance to more complex transformations, such as non-linear or higher-dimensional transformations, may require different analyses, limiting the applicability of the results to a broader set of machine learning problems.

**Questions:**

* The paper focuses on deep linear networks for tractability, but how do you anticipate the results extending to non-linear neural networks, which are more prevalent in practical applications? Have you considered the potential challenges or modifications needed for such generalization?

* How do you expect the optimization landscape and critical points to change when considering more complex or non-standard invariance structures? Could your theoretical framework be adapted to handle these?

* Given the computational efficiency noted in your experiments, how scalable are the findings, particularly regarding the comparison between data augmentation and hard-wiring, when applied to much larger models or datasets, such as in convolutional or transformer networks?

---

> ### Author Response · Authors · 2024-11-20
>
> We appreciate the reviewer’s comments and feedback.
>
> **1. Regarding the limitation to linear networks:**
>
> We consider a model of rank constrained linear maps. While this is indeed a relatively simple model, one of its interesting properties is that it has a nonlinear function space. Part of our motivation to study this model is that it could inform us about how invariance constraints might interact with a nonlinear function space. Networks with nonlinear activation functions are typically also nonlinear. We think that rank bounded linear functions are an interesting and natural place to start, but agree that studying other types of constraints will be important towards obtaining a more complete picture of general invariant neural networks. Linear convolutional networks might be an interesting model to consider next in conjunction with invariance, which in contrast to fully connected linear networks may have spurious local minima and richer types of constraints in function space.
>
> **2. Regarding the limitation to invariance in the linear context:**
>
> It is interesting to consider more general types of invariances. Most group invariances people use in practice are linear, such as rotation and translation. In representation theory, a representation is defined as a group homomorphism between $\mathcal{G}$ and $GL(V)$, i.e., $\rho: \mathcal{G} \rightarrow GL(V)$. Thus, when people set up the framework for invariant or equivariant models, they implicitly assume that the transformations are linear.
>
> **3. Regarding other architectures:**
>
> We agree that linear networks are a relatively simple model. One of the interesting aspects of our work is that we consider function spaces that are not required to be linear. This means that some phenomena and conclusions might be transferrable to other architectures. We suggest that data-augmented model may have similar performance to hard-wired ones, though at the cost of data and computing efficiency. Regularized models do not require more data and should have close performance to the hard-wired model, but may induce more critical points than the other two methods. Hard-wired model should have the best performance, though one might need to design the invariant architecture carefully before feeding the data to the model. These propositions of course will necessitate further investigation, and we hope our work might serve as an inspiration or starting point.

---

> ### Author Response · Authors · 2024-11-22
>
> Dear Reviewer,
>
> Thank you for your time and feedback on our paper. We have provided a detailed response addressing your concerns. Since then, we have not seen any follow-up comments from you. If there are additional questions or concerns that remain unresolved, we would be happy to address them further. Your feedback is valuable to us, and we sincerely hope to help clarify any lingering doubts.

---

> ### Comment · Reviewer_GZqx · 2024-11-24
> **Re:**
>
> Thank you to the authors for their response. However, I would like to maintain my current score, as focusing on linear networks substantially limits the contributions of this paper.

---

> > ### Author Response · Authors · 2024-11-27
> >
> > Thank you for your response. We have updated the manuscript to include an appendix for experiments in nonlinear networks in case of any interest. Empirical results in Appendix A.11 indicate that data augmentation and a constrained model indeed achieve a similar loss in the late phase of training for two-layer neural networks with different activation functions. At the same time, we observe that for models with a higher expressive power, it is more difficult to learn invariance from the data. These preliminary empirical results may give us some intuition to study the case of nonlinear models theoretically.

---

### Official Review · Reviewer_En8P · 2024-11-03

**Soundness:** 3
**Presentation:** 3
**Contribution:** 3
**Rating:** 6
**Confidence:** 4

**Summary:**

This paper explores the loss landscape of three approaches to achieve invariance: data augmentation, network hard-wiring, and regularization. The authors solve the optimization problems arising from each method and find that the first two approaches share the same critical points, whereas regularization leads to a larger number of critical points. Additionally, experiments show that data augmentation indeed results in invariance of the trained network, and that both data augmentation and hard-wiring converge to the same global optimum.

**Strengths:**

- The authors theoretically solve the optimization problems arising from three different approaches in a bounded-rank linear model and discuss their solutions. This analysis offers fresh insights for the learning-with-symmetry community on the use and comparison of methods to achieve invariance.
- They not only characterize the global optima of these approaches but also identify all the critical points, demonstrating that regularization leads to a greater number of critical points, which is an interesting result.
- The authors verify their theoretical findings through experiments on the rotated MNIST dataset, showing that while both hard-wiring and data augmentation converge to the same global optimum, data augmentation demands higher computational costs.
- The paper is well-written.

**Weaknesses:**

- As noted on line 169, the theoretical scope of this paper is limited to finite and cyclic groups. However, in many real-world applications, the relevant groups are not finite or cyclic, such as permutation group [1], rotation group [2], and sign and basis transformation group [3]. This limitation reduces the practical applicability of the paper’s findings.
- Some assumptions in the paper lack adequate justification. For instance, in Remark 2, the authors use $Y=WX+E$ to suggest that the rank assumption of $\overline{Z}^\mathrm{inv}$ is mild. However, the noise matrix in this example seems unrealistic, as real datasets typically have structured rather than random correlations between data and labels. In Corollary 1, the authors assume that the singular values of three matrices are pairwise distinct, but this assumption is not justified. Verifying whether these assumptions hold in real datasets would improve the paper’s applicability.
- Some key citations are missing. In Proposition 1, the authors characterize invariant linear maps under a cyclic group. However, [1] previously characterized all invariant and equivariant linear maps for symmetric groups, and [4] extended this work to identify all polynomials equivariant under symmetric groups.

[1] Maron, H., Ben-Hamu, H., Shamir, N., & Lipman, Y. (2018, September). Invariant and Equivariant Graph Networks. In International Conference on Learning Representations.

[2] Dym, N., & Maron, H. On the Universality of Rotation Equivariant Point Cloud Networks. In International Conference on Learning Representations.

[3] Ma, G., Wang, Y., Lim, D., Jegelka, S., & Wang, Y. (2024). A Canonicalization Perspective on Invariant and Equivariant Learning. arXiv preprint arXiv:2405.18378.

[4] Puny, O., Lim, D., Kiani, B., Maron, H., & Lipman, Y. (2023, July). Equivariant Polynomials for Graph Neural Networks. In International Conference on Machine Learning (pp. 28191-28222). PMLR.

**Questions:**

In Figure 3(a), why doesn’t the non-invariant component of $W$ converge to zero? Additionally, in Figure 4(a), why does the non-invariant component of $W$ increase for data augmentation?

---

> ### Author Response · Authors · 2024-11-20
>
> We appreciate the reviewer’s insightful comments and feedback.
>
> **1. Regarding the limitation of cyclic groups or finitely generated groups:**
> We presented our results for the case of cyclic groups or finitely generated groups. However, this is not a serious limitation in the following sense. Extending our results to continuous groups (such as $SO(n)$ and $SE(n)$) is straightforward, since Lie theory provides a way of analyzing continuous groups in terms of their infinitesimal generators. To explain this point, we have updated the manuscript to include Remark 1 and an Appendix A.1 with details about the extension of our results to Lie groups.
>
> **2. Regarding the assumptions in Remark 2:**
>
> It is true that the noise can be structured in real data. In our assumption, we only need that the noises for different data entries are independent and identically distributed. While simplistic, this is a relatively common assumption and is relatively mild in practice. We added an appendix where we empirically computed the spectrum of target matrices in MNIST and verified that they satisfy the assumptions.
>
> **3. Regarding citations:**
>
> Thank you for providing these relevant references. We have updated our manuscript to include them.
>
> **4. Regarding figure 3 and figure 5 (originally figure 4):**
>
> In Figure 3(a), we are plotting the Frobenius norm of the non-invariant component of the end-to-end matrix, i.e., $||W^{\perp}||_F$, for data augmentation and regularization trained with mean squared loss. For data augmentation, it actually converges to zero if we allow more epochs. For regularization, since the penalty coefficient $\lambda$ is finite, the critical points are actually not invariant. Therefore, in those cases $||W^{\perp}||_F$ does not converge to zero. However, the larger the regularization strength, the smaller the Frobenius norm of the non-invariant component at convergence, in line with the theory. In Figure 5(a), we again are plotting $||W^{\perp}||_F$ for data augmentation and regularization trained on the same dataset, but with cross entropy loss. It is observed that, for larger $\lambda$, the dynamics of $||W^{\perp}||_F$ resemble those when trained with MSE. On the other hand, for small $\lambda$, $||W^{\perp}||_F$ may increase at first, and then decrease. For data augmentation, we observe that $||W^{\perp}||_F$ actually decreases after increasing if we allow more epochs. Our theoretical results only support the scenario for mean squared loss. Thus, when trained with cross entropy, we cannot say whether all the critical points are invariant or not. Further research needs to be done to investigate this.

---

> > ### Comment · Reviewer_En8P · 2024-11-22
> > **Response to Authors**
> >
> > Thank you for your response and for providing additional discussion on the assumptions, along with the reference to Proposition 5 from Finzi et al. I appreciate the clarification and will be maintaining my score.

---

### Official Review · Reviewer_QRkn · 2024-11-04

**Soundness:** 3
**Presentation:** 2
**Contribution:** 2
**Rating:** 6
**Confidence:** 3

**Summary:**

The paper explores the loss landscape geometry of invariant deep linear neural networks, focusing on three approaches for enforcing invariance: data augmentation, regularization, and hard-wiring. The paper provides a theoretical comparison, demonstrating that the global optima for all three methods are equivalent in the limit of strong regularization and full data augmentation. It also examines the critical points of the loss landscapes, showing that data augmentation and hard-wiring result in identical sets of critical points, while regularization introduces additional ones. Empirical experiments show that training with data augmentation converges to a critical point that parametrizes an invariant function, data augmentation is computationally more expensive than hard-wiring, and regularization falls in between.

**Strengths:**

- The paper provides a mathematically rigorous analysis of the loss landscapes for the three approaches and is a good contribution to the study of loss landscapes.
- By establishing that data augmentation and hard-wiring result in identical critical points, the paper offers a unifying perspective on invariance in optimization. This connection complements recent study on comparing equivariant architectures and data augmentation.
- The empirical results align well with the theoretical findings, providing concrete evidence that training with data augmentation converges to a critical point that parametrizes an invariant function.

**Weaknesses:**

The settings considered in the paper seem limited. In particular:
- As the authors also acknowledge, this paper focuses on deep linear networks. The results depend heavily on properties almost unique to this type of networks, such as that the network’s function space is a vector space of linear maps. There does not seem to a clear path that could extend the results here to other, especially nonlinear, architectures.
- The main results are limited to cyclic and finitely generated groups, which does not apply to continuous groups in common datasets, such as rotation and scaling.

**Questions:**

- I was not able to follow Section 4.3 and would appreciate any clarification on the main conclusion from the experiment or which theorem the experiment seeks to support.
- Nordenfors et al. (2024) points out that the set of stationary points are identical for data-augmentation and hard-wiring, on both linear and certain nonlinear architectures. Could the authors comment on whether these results are more general than Theorem 3?
- Can the results in this paper provide useful insights for practitioners? I do not believe a lack of immediate practical implication is a major weakness, but the paper might reach more audience by including some motivation for studying the loss landscape or invariant deep linear networks.

---

> ### Author Response · Authors · 2024-11-20
>
> We appreciate the reviewer’s insightful comments and feedback.
>
> **1. Regarding your comment about the settings being limited to vector spaces:**
>
> Kindly observe that the model we are considering consists of functions that are linear over the input variable but that the function space itself is not a vector space. Our function space consists of linear maps with bounded rank. In general this is a so-called determinantal variety and is not a convex set. It is a vector space only in the special case where the bound on the rank is trivial (i.e., equal to either the number of rows or columns). Due to the non-convexity of the function space, the effect of the constraints imposed by the invariances is not obvious. We regard this as one of the most interesting parts of our work. We consider this setting precisely to gain a better understanding of  function spaces that are nonlinear subsets of vector spaces, which is also the case in networks with nonlinear activation functions.
>
> **2. Regarding your comment about the settings being limited to cyclic and finitely generated groups:**
>
> We presented our results for the case of cyclic groups or finitely generated groups. However, we contend that this is not a serious limitation. Extending our results to continuous groups (such as $SO(n)$ and $SE(n)$) is straightforward, since Lie theory provides a way of analyzing continuous groups in terms of their infinitesimal generators. To explain this, we have updated the manuscript to include Remark 1 and an Appendix A.1 with details about the extension of our results to Lie groups.
>
> **3. Regarding the comparison with Nordenfors et al., (2024) [1]:**
>
> Regarding Theorem 3.7 in the work of Nordenfors et al., 2024 [1]:
> 1) That result shows that the data augmented model and the hard-wired model have the same stationary points within the set of representable equivariant maps $\mathcal{E}$. As they state in their Remark 3.8.2, the set $\mathcal{E}$ is assumed to be a vector space. This means that the result considers only a limited class of architectures. In our work, in contrast, the function space is not required to be a vector space. Our function space is a vector space only in the special case where the rank constraint is trivial. The fact that our function space is not required to be a linear space is in our opinion one of the most interesting aspects of our work.
> 2) Furthermore, as stated in Nordenfors' Remark 3.8.1, their theorem only applies to points in $\mathcal{E}$. This means that their result does not say anything about stationary points of the data augmented model that are not in $\mathcal{E}$. In contrast, our results describe all critical points for a non-linear rank-constrained function space and we show that all of them are indeed invariant. Finally, we also compare the regularized model with the other two, whereas Nordenfors et al. do not consider regularization in their work.
>
> **4. Regarding Section 4.3:**
>
> The experiments in Section 4.3 show the training dynamics for linear networks under the settings that we consider. Since the parametrized function $\hat{W}$ is linear, we can decompose it orthogonally into two parts, a non-invariant part $\hat{W}^{\perp}$ and an invariant part $\hat{W}-\hat{W}^{\perp}$. We are plotting the Frobenius norm of both components during training. This is to monitor whether the non-invariant part converges to zero and at what rate for the case of data augmentation and regularization.
> The experiments show that as the regularization strength increases, the result of training will tend to have a smaller non-invariant component. This is in line with our theory in Theorem 2. Our theoretical results focus on the static loss landscape rather than training dynamics. The experiments section complements this experimentally and hints at interesting phenomena that merit further investigation. We have updated the manuscript for better illustration.
>
>
> **5. Useful insights for practitioners:**
>
> Though our theoretical results are for linear networks with non-convex function space, it is natural to conjecture that some phenomena might carry over to other overparametrized models with non-convex function space. We suggest that the data-augmented model may have similar performance to hard-wired ones, but will incur higher data and computing costs. The regularized model does not require more data and should have a performance close to the hard-wired model, but it may induce more critical points than the other two methods. The hard-wired model should have the best performance, though one might need to design the invariant architecture carefully before feeding the data to the model.
>
>
> References:
>
> [1] Oskar Nordenfors, Fredrik Ohlsson, and Axel Flinth. Optimization dynamics of equivariant and augmented neural networks, 2024.

---

> > ### Comment · Reviewer_QRkn · 2024-11-23
> >
> > I appreciate the authors’ clarifications on the limitations and detailed explanations on related work and experiments. I maintain my positive rating.

---

### Official Review · Reviewer_wcVq · 2024-11-04

**Soundness:** 2
**Presentation:** 2
**Contribution:** 1
**Rating:** 6
**Confidence:** 2

**Summary:**

The paper studies the three related problems of learning the linear predictor under squared loss. The problem is not simple since the output dimension is $>1$ and it has been known since the seminal work of Baldi & Hornik that there are small-rank optimal matrices that generate saddles in the original problem. The paper studies similar problems in the data augmentation and regularization cases and shows the same loss landscape characteristics.


------------------------------------------------------------------------

After the author's response, I increased my score to 5. I may increase again during the discussion phase between the reviewers.


------------------------------------------------------------------------

After the author's second response, I increased my score to 6.

**Strengths:**

Comparing learning with data augmentation vs an invariant architecture is an interesting problem and linear networks may be a good place to start for a theoretical study. The paper is written rather well but the global structure of the paper (presentation order) is confusing (see Weaknesses).

**Weaknesses:**

It is not clear how much technical novelty there is in the paper. Proposition 2 is copied from Zhang and Theorem 4 is copied from Trager. Theorem 4 seems rather trivial, how is this novel compared to Baldi and Hornik?

The landscape complexity questions are interesting for complex losses where the full set of critical points is not known. In this paper, all local minima are global which is a stronger result on the loss landscape (and yes, possible for linear networks). Interestingly, the global minima are equivalent between problems in the infinite data limit, but what happens for finite data? Is there a separation between the problems? Like using invariance is better than using data augmentation? That kind of result would make the paper much more interesting. I might have missed such a point in the paper due to quick reading and confusing organization of the paper.
I think it'd be better to state the three problems early in the paper similar to

Levin, Eitan, Joe Kileel, and Nicolas Boumal. "The effect of smooth parametrizations on nonconvex optimization landscapes." Mathematical Programming (2024): 1-49.

**Questions:**

See weaknesses.

---

> ### Author Response · Authors · 2024-11-20
>
> We appreciate the reviewer’s comments and feedback.
>
> **1.  Regarding novelty and differences from previous literatures:**
>
> As indicated in the introduction, Baldi \& Hornik (1989) [1] studied two-layer linear networks with no constraints on the weight matrices other than their format and trained with the square loss. They show that this optimization problem has single minimum up to a trivial symmetry and all other critical points are saddles. Notice that this general structure fundamentally results from the properties of low-rank matrix approximation with the square loss and corresponding results from the 1930s. That work does not investigate learning invariances in the data. In contrast, we aim to study how data augmentation, regularization in function space, and imposing constraints based on invariances can affect the optimization problem and the loss landscape. We specifically consider a function space subject to rank constraints, which is a determinantal variety and not a vector space, and for which it is not clear how it interplays with the constraints imposed by invariance. We are not aware of previous works studying this setting.
>
> **2. Regarding Proposition 2 and Proposition 7(originally Theorem 4):**
> 1) For Proposition 2:
> We obtain this by applying manifold regularization results from Zhang \& Zhao (2013) [3], Theorem 1. The difference is that the symmetric positive semidefinite matrix that characterizes the low-dimensional structure of the manifold behind the weight matrix is $\tilde{G}\tilde{G}^T$ in our context.
> 2) For Proposition 7:
> We had included a version of Trager's Theorem 28 for clarity of presentation. We have now moved it to the appendix. Observe that this result is a version of results that have appeared in diverse works on low rank matrix approximation following from Mirski's characterization. A minor difference is that we included the description of the critical points for the case of low rank targets, which was not included by Trager et al. (2020) [4].
>
> **3. Regarding your question about finite data:**
>
> In our paper, we don't have any assumptions related to infinite data. All the results hold for a finite amount of data in general position. We are able to prove that the critical points in function space for data augmentation and hard-wired invariant model are the same, implying that the generalization performance is be the same for both methods if global optima are achieved.
>
> **4. Regarding the comparison with Levin et al. (2024) [2]:**
>
> We are familiar with the work of Levin et al. (2024) [2], which studies the effect that the parametrization map has on an optimization landscape and specifically establishes conditions for when a critical point in the parameter space corresponds to a critical point in the function space. In contrast, we aim to characterize the global optima in a rank-constrained function space for the optimization problem in data augmentation, regularization, and hard-wired models. Discussing the effect of parametrization in general is not a main focus of our paper. Nonetheless, we can show that there are no local minima other than global minima in the parameter space when the model is parametrized as the product of linear weight matrices.
>
>
> References:
>
> [1] P. Baldi and K. Hornik. Neural networks and principal component analysis: Learning from examples without local minima. Neural Networks, 2(1):53–58, 1989.
>
> [2] Eitan Levin, Joe Kileel, and Nicolas Boumal. The effect of smooth parametrizations on nonconvex optimization landscapes. Math. Program., March 2024. ISSN 0025-5610, 1436-4646. doi: 10.1007/s10107-024-02058-3.
>
> [3] Zhenyue Zhang and Keke Zhao. Low-rank matrix approximation with manifold regularization. IEEE Transactions on Pattern Analysis and Machine Intelligence, 35(7):1717–1729, 2013. doi: 10.1109/TPAMI.2012.274.
>
> [4] Matthew Trager, Kathlen Kohn, and Joan Bruna. Pure and spurious critical points: a geometric study of linear networks. In International Conference on Learning Representations, 2020.

---

> > ### Comment · Reviewer_wcVq · 2024-11-25
> > **thank you for the response**
> >
> > Thank you for the clarifications. I re-read the experimental section. Together with your response and re-reading this part, I now understand better where your paper stands in the literature. I'm increasing my score to 5. I may increase again during the discussion phase between the reviewers.
> >
> > Regarding the literature on how overparameterization facilitates optimization, I believe a missing citation is
> >
> > Simsek, Berfin, et al. "Geometry of the loss landscape in overparameterized neural networks: Symmetries and invariances." International Conference on Machine Learning. PMLR, 2021.
> >
> > The above paper studied the non-convexity of the non-linear neural networks from the landscape complexity point of view. See also
> >
> > Simsek, Berfin, et al. "Should Under-parameterized Student Networks Copy or Average Teacher Weights?." Advances in Neural Information Processing Systems 36 (2024).
> >
> > for a characterization of the critical points of the neural network loss with MSE loss, when using the Gaussian error activation function.

---

> > > ### Author Response · Authors · 2024-11-27
> > > **Thank you for your response.**
> > >
> > > Thank you for re-reading our paper as well as the responses. We have updated the manuscript to include these literatures you mentioned in the introduction section. We have also included an appendix for experiments in nonlinear networks in case of any interest.  Empirical results in Appendix A.11 indicate that data augmentation and a constrained model indeed achieve a similar loss in the late phase of training for two-layer neural networks with different activation functions. At the same time, we observe that for models with a higher expressive power, it is more difficult to learn invariance from the data.

---

> ### Author Response · Authors · 2024-11-22
>
> Dear Reviewer,
>
> Thank you for your time and feedback on our paper. We have provided a detailed response addressing your concerns. Since then, we have not seen any follow-up comments from you. If there are additional questions or concerns that remain unresolved, we would be happy to address them further. Your feedback is valuable to us, and we sincerely hope to help clarify any lingering doubts.

---

### Author Response · Authors · 2024-12-03
**Revision Summary**

Dear reviewers and AC,

We sincerely appreciate your valuable time and effort spent reviewing our manuscript. As highlighted by the reviewers (QRkn, En8P), we provide a mathematically rigorous analysis of the loss landscapes for the three approaches to achieve an invariant estimator (data augmentation, hard-wiring, and regularization). Meanwhile, as GZqx pointed out, our theory only supports linear networks with bottleneck layers. Thus, we added an appendix (A.11) to include experiments for networks with nonlinear activation functions.

We appreciate your constructive feedback on our manuscript. In response to the comments, we have carefully revised and enhanced the manuscript as follows:
* We updated the manuscript to include more references about related works, as suggested by the reviewers.
* We added an appendix (A.1) to discuss the extension to continuous groups based on Lie theory.
* We added an appendix (A.11)  for experiments in nonlinear networks in case of any interest. Preliminary results show data augmentation and a constrained model indeed achieve a similar loss in the late phase of training for two-layer neural networks with different activation functions. At the same time, we observe that for models with a higher expressive power, it is more difficult to learn invariance from the data.
* We added an appendix (A.9) to justify that the assumptions in our theory are satisfied on real-world datasets (e.g., MNIST).
* We reorganized the related work section in the introduction (Section 1.2) to better discuss related works and illustrate our contributions.
* We rewrote the experiment section 4.3 to clarify our claims.

Thank you very much,

Authors.

---

### Meta-Review · Area_Chair_C8Gd · 2024-12-05

**Metareview:**

The paper explores the effects of different methods for enforcing invariance, hard-wiring, data augmentation, and regularization, on the optimization landscapes associated with deep linear networks. The reviewers found the general conceptual approach appealing. However, opinions were somewhat divided on whether the choice of model (deep linear networks) and groups (cyclic and finitely generated) serves as a reasonable starting point for theoretical investigation or whether the gap from practical scenarios is too wide to make the findings meaningful and applicable. Despite considerable discussions with the authors, the overall evaluation remained lukewarm, with doubts persisting about the relevance of these results to real-world models. The authors are encouraged to incorporate the important feedback given by the knowledgeable reviewers.

**Additional Comments On Reviewer Discussion:**

See above.

Rather, I'll use this box for a general comment. The reviewers and this AC found the general question studied: how main approaches to incorporating invariance differ, intriguing. However, the paper does not seem to have effectively convinced the audience why the model chosen might be a fair starting point (among other concerns).

---

### Decision · Program_Chairs · 2025-01-22

Reject